# Benchmarks for Physical Reasoning AI

**Andrew Melnik**[*]                                   *andrew.melnik.papers@gmail.com*
*Center for Cognitive Interaction Technology, University of Bielefeld, Germany*

**Robin Schiewer**[*]                                   *robin.schiewer@ini.rub.de*
*Institute for Neural Computation, Department of Computer Science*
*Ruhr University Bochum, Germany*

**Moritz Lange**[*]                                   *moritz.lange@ini.rub.de*
*Institute for Neural Computation, Department of Computer Science*
*Ruhr University Bochum, Germany*

**Andrei Muresanu**                                   *andrei.muresanu@uwaterloo.ca*
*University of Waterloo, Canada*
*Vector Institute, Canada*

**Mozhgan Saeidi**                                   *mozhgans@stanford.edu*
*Department of Computer Science, University of Toronto, Canada*
*Vector Institute, Canada*
*Department of Computer Science, Stanford University, USA*

**Animesh Garg**                                   *animesh.garg@gatech.edu*
*University of Toronto, Canada*
*Vector Institute, Canada*
*Georgia Institute of Technology, USA*

**Helge Ritter**                                   *helge@techfak.uni-bielefeld.de*
*Center for Cognitive Interaction Technology, University of Bielefeld, Germany*

**Reviewed on OpenReview:** *https://openreview.net/forum?id=cHroS8VIyN*

*Awesome* **list:** *https://github.com/ndrwmlnk/Awesome-Benchmarks-for-Physical-Reasoning-AI*

## Abstract

Physical reasoning is a crucial aspect in the development of general AI systems, given that human learning starts with interacting with the physical world before progressing to more complex concepts. Although researchers have studied and assessed the physical reasoning of AI approaches through various specific benchmarks, there is no comprehensive approach to evaluating and measuring progress. Therefore, we aim to offer an overview of existing benchmarks and their solution approaches and propose a unified perspective for measuring the physical reasoning capacity of AI systems. We select benchmarks that are designed to test algorithmic performance in physical reasoning tasks. While each of the selected benchmarks poses a unique challenge, their ensemble provides a comprehensive proving ground for an AI generalist agent with a measurable skill level for various physical reasoning concepts. This gives an advantage to such an ensemble of benchmarks over other holistic benchmarks that aim to simulate the real world by intertwining its complexity and many concepts. We group the presented set of physical reasoning benchmarks into subcategories so that more narrow generalist AI agents can be tested first on these groups.

---

[*]Shared first authorship

# 1 Introduction

Physical reasoning refers to the ability of an AI system to understand and reason about the physical properties and interactions of objects. While traditional machine learning models excel at recognizing patterns and making predictions based on large amounts of data, they often lack an inherent understanding of the underlying physical mechanisms governing those patterns. Physical reasoning aims to bridge this gap by incorporating physical laws, constraints, and intuitive understanding into the learning process.

In order to assess, contrast, and direct AI research efforts, benchmarks are indispensable. When starting a new study, it is essential to know the landscape of available benchmarks. Here we propose such an overview and analysis of benchmarks to support researchers interested in the problem of physical reasoning AI. We provide a comparison of physical reasoning benchmarks for testing deep learning approaches, identify groups of benchmarks covering basic physical reasoning aspects, and discuss differences between benchmarks.

Table 1: Benchmarks and their reasoning tasks: descriptive (D), predictive (P), explanatory (E) and counterfactual (C), defined in Yi et al. (2020) and explained in Section 2.11. While interactive benchmarks allow agents to act in the world, passive benchmarks merely provide observational data. Each benchmark name links to its website or repository.

|      | Benchmark | Agent | Reasoning Task |
|------|-----------|-------|----------------|
| 2.1  | PHYRE | Interactive | Make objects touch |
| 2.2  | Virtual Tools | Interactive | Make objects touch |
| 2.3  | Phy-Q | Interactive | Hit objects with slingshot |
| 2.4  | Physical Bongard Prob. | Passive | Recognize conceptual differences (D) |
| 2.5  | CRAFT | Passive | Answer questions (D, E, C) |
| 2.6  | ShapeStacks | Passive | Stability prediction (P) |
| 2.7  | SPACE | Passive | Recogn. containment, interaction (D), or future frame prediction (P) |
| 2.8  | CoPhy | Passive | Predict future state from changed initial state (P, C) |
| 2.9  | IntPhys | Passive | Judge physical feasibility (D) |
| 2.10 | CATER | Passive | Recognize compositions of object movements (D) |
| 2.11 | CLEVRER | Passive | Answer questions (D, P, E, C) |
| 2.12 | ComPhy | Passive | Answer questions (D, P, C) |
| 2.13 | CRIPP-VQA | Passive | Answer questions (D, C), Planning |
| 2.14 | Physion | Passive | Predict object contact (P) |
| 2.15 | Language-Table | Interactive | Push objects to absolute or relative positions |
| 2.16 | OPEn | Interactive | Push objects to relative positions |

Reasoning about physical object interactions and physical causal effects may be one of the grounding properties of human-level intelligence that ensures generalization to other domains. The ability to adapt to an unknown task across a wide range of related tasks, known as a broad generalization, is gaining increasing attention within the AI research community (Malato et al., 2023; Milani et al., 2023; Malato et al., 2022). We propose to use a set of specialized benchmarks to test generalist physical reasoning AI architectures.

In this survey, we discuss 16 datasets and benchmarks (see Tables 1 and 2) to train and evaluate the physical reasoning capabilities of AI agents. The benchmarks we examine here involve a range of physical variables which are central to physical interactions amongst material objects, such as size, position, velocity, direction of movement, force and contact, mass, acceleration and gravity, and, in some cases, even electrical charge. The observability of these variables is strongly affected by the perceptual modalities (e.g. vision, touch) that are available to an agent. Such representations in the form of empirical rules can be perfectly successful for providing an intuitive physical understanding whose development in humans and animals has itself been a subject of research (Melnik et al., 2018).

Table 2: Benchmarks categorized according to their concepts, scene composition, and physical variable types: global (G), immediate (I), temporal (T), and relational (R) (see Section 1). If relational variables are not explicitly stated this means that relevant relational variables are correlated with (and therefore become) immediate variables. An example of this would be if object mass is directly correlated with object size. Physical concepts are, to some extent, a question of abstraction (e.g. bouncing implies collision and deformation) so the concepts column here summarizes general and somewhat orthogonal base concepts.

|      | Benchmark | Concepts | Variables | Scene |
|------|-----------|----------|-----------|-------|
| 2.1  | PHYRE | Collision, Falling | G, I, T | 2D simplistic |
| 2.2  | Virtual Tools | Collision, Falling | G, I, T | 2D simplistic |
| 2.3  | Phy-Q | Collision, Falling | G, I, T | 2D realistic |
| 2.4  | Physical Bongard Problems | Collision, Falling, Containment | G, I | 2D simplistic |
| 2.5  | CRAFT | Collision, Falling | G, I, T | 2D simplistic |
| 2.6  | ShapeStacks | Falling, Stacking | G, I | 3D realistic |
| 2.7  | SPACE | Collision, Falling, Containment, Occlusion | G, I, T | 3D simplistic |
| 2.8  | CoPhy | Collision, Falling, Stacking | G, I, T, R | 3D realistic |
| 2.9  | IntPhys | Occlusion | G, I, T | 3D realistic |
| 2.10 | CATER | Containment, Occlusion, Lifting | G, I, T | 3D simplistic |
| 2.11 | CLEVRER | Collision | G, I, T | 3D simplistic |
| 2.12 | ComPhy | Collision, Attraction / Repulsion | G, I, T, R | 3D simplistic |
| 2.13 | CRIPP-VQA | Collision | G, I, T | 3D simplistic |
| 2.14 | Physion | Coll., Falling, Containment, Stacking, Draping | G, I, T | 3D realistic |
| 2.15 | Language-Table | Collision | G, I | 3D simplistic |
| 2.16 | OPEn | Collision | G, I, T | 3D simplistic |

To judge the complexity of benchmarks for machine learning approaches, we divide physical variables that belong to individual objects, according to their observability and accessibility, into three types:

1. **Immediate** object variables visible from single frames (e.g. size, position)

2. **Temporal** object variables exposed through time evolution (e.g. velocity, direction of movement)

3. **Relational** object variables exposed only through object interaction (e.g. charge, mass)

In addition to these object variables, there are also global environment variables which affect all objects equally, such as gravity.

Object variables become harder to extract the less obvious they are, i.e. the more complex an observation has to be in order to extract a variable. For immediate variables, it suffices to understand an image. Temporal variables already require understanding a video and relational variables, furthermore, require to relate objects to each other within a video.

The benchmarks we review do all contain global environment variables and immediate object variables. However, only a subset considers temporal variables and only very few, finally, cover truly relational variables. While many benchmarks contain variables such as mass, these are often effectively transformed from relational into immediate variables when they become obvious from individual images, e.g. when object mass is directly correlated with object size.

The benchmarks in this survey consider classical mechanics. Benchmarks that require (scientific and/or intuitive) understanding of phenomena related to other branches of physics, such as hydrodynamics, optics, or even quantum mechanics, are out of the scope of this survey. Likewise, pure question-answering benchmarks that don't require any physical understanding are also out of the scope of this survey. Many robot movements, such as walking, climbing, jumping, or throwing, could benefit strongly from some intuitive physical understanding to be executable successfully and safely. However, robotics benchmarks often place more of a focus on competencies such as combination of embodiment König et al. (2018) and control Sheikh et al. (2023), perception, navigation, manipulation, and localization, with physical reasoning simply being a potentially useful property. Therefore, we've decided to exclude most robotics benchmarks from this paper.

In Section 2 we describe the details of each benchmark, including solutions, related work and a discussion of its contribution to the landscape of physical reasoning benchmarks. After that, we provide a grouping of benchmarks according to physical reasoning capabilities in Section 3. Section 4 gives a brief introduction to physical reasoning approaches based on the approaches proposed for the various benchmarks. Section 5 concludes the paper with a discussion.

## 2 Physical Reasoning Benchmarks

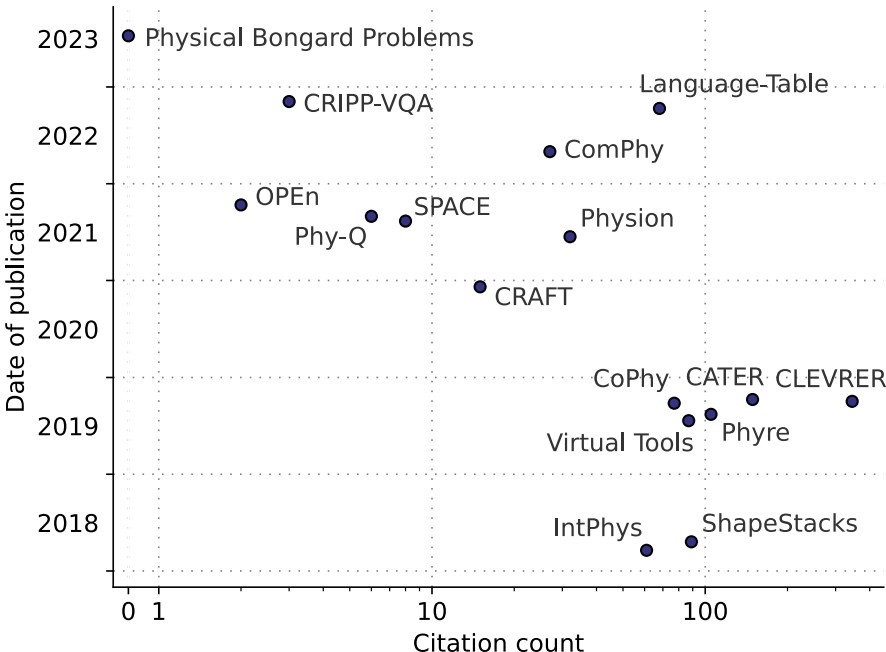

Figure 1: Publication date versus citations (symmetric log scale) for benchmarks. The date of publication is based on the first submission on arXiv or elsewhere. Citation count is a snapshot of Google Scholar information from 7 December 2023.

This section presents each benchmark in detail. Each description is separated into a general introduction, solution and related work, and finally the contribution of this benchmark. The general description contains an overview of the scope, scene, reasoning task, and potentially other relevant aspects of a benchmark. In solution and related work we list baseline solutions, further solutions, and potentially other related work. In contribution, finally, we provide a view on how this benchmark fits into the landscape of physical reasoning benchmarks.

In addition to this, we summarize information about scene, reasoning task, physical concepts and technical data formats, all at a glance, in Tables 1, 2 and 3, respectively. In particular, the first two tables contain links to repositories that are outdated in some original papers and have been updated for this publication. The technical data in Table 3 is often only available when downloading and analysing repositories, which makes this table the only easily accessible source of such information. In order to judge a benchmark's reception, Figure 1 details the publication history and number of citations as a proxy measure for relevance in the field.

We have established a dynamic list of physical reasoning benchmarks[1] that can be continuously enhanced through the submission of pull requests for new benchmarks.

---

[1] https://github.com/ndrwmlnk/Awesome-Benchmarks-for-Physical-Reasoning-AI

## 2.1 PHYRE

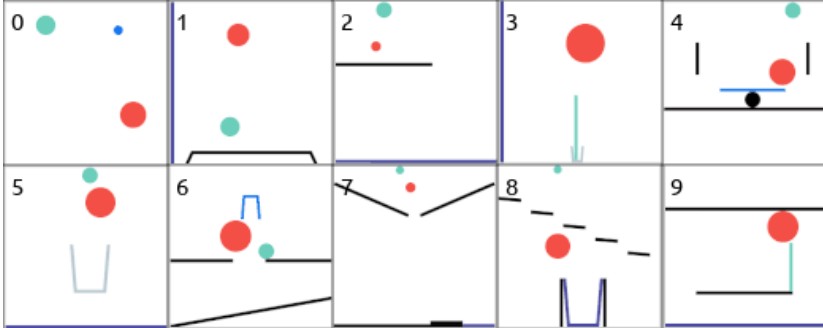

Figure 2: Ten templates of PHYRE-B (BALL) puzzles (Bakhtin et al., 2019). Task: Place the red action balls so that a green object ends up touching a blue surface. Image adapted from (Harter et al., 2020).

PHYRE (Bakhtin et al., 2019) studies physical understanding based on visual analysis of a scene. It requires an agent to generate a single action that can solve a presented puzzle. The benchmark is designed to encourage the development of learning algorithms that are sample-efficient and generalize well across seen and unseen puzzles.

Each of the puzzles in PHYRE is a deterministic 2D box-shaped environment following Newtonian physics. It is comprised of a goal and a set of rigid objects that are either static or dynamic, whereas only the latter can be influenced by gravity or collisions. To indicate their object category, dynamic objects can have various colors, and static objects are always black. The goal condition of a puzzle refers to two objects that should touch each other for at least 3 seconds and is not available to the agent. Observations are provided in the form of images, and valid actions are the initial placement of one or two red balls and the choice of their radii. Note that only a single action in the first frame is allowed; from there on, the simulation unfolds without any interference by the agent.

PHYRE contains two benchmark tiers: *Phyre-B*, which requires choosing the position and radius of one red ball, Fig 2, and *Phyre-2B*, which requires choosing the same attributes for two balls. To limit the action search space, both tiers come with a predefined set of 10000 action candidates. Each tier contains 25 templates from which 100 concrete tasks per template are produced. While a template only defines the set of objects and the scene goal, tasks fill in the detailed object properties such as position and size. Since tasks from a single template are more similar than those from different templates, PHYRE differentiates between the within and the cross-template benchmark scenarios. Specifically, the cross-template scenario is intended to test the generalization capabilities of an agent beyond known object sets. At test time, PHYRE requires the agent to solve a task with as few trials as possible, whereas each attempt results in a binary reward that indicates whether the puzzle was solved or not. This provides the opportunity to adapt and improve the action of choice in case of a failed attempt. Two measures are taken to characterize the performance of an agent:

- The *success percentage* is the cumulative percentage of solved tasks as a function of the attempts per task.

- To put more emphasis on solving tasks with few attempts, the *AUCCESS* is a weighted average of the success percentages $s_k$ computed as $\sum_k w_k \cdot s_k / \sum_k w_k$ with $w_k = \log(k+1) - \log(k)$ for the number of attempts $k \in \{1, ..., 100\}$. Note that this results in an AUCCESS of less than 50% for agents that require more than 10 attempts on average to solve tasks.

**Solutions and related work** The baseline solution to PHYRE is based on the idea to learn a critic-value function for state-action pairs, where the state is an embedding of the initial frame of an episode, and the action is a specific choice of the red ball's position and radius. Then, a grid search over a predefined set of 3-dimensional actions for a given state is performed, and actions are ranked w.r.t. the value estimated by

the *critic* network. The set of top-ranked candidate actions by the *critic* network is provided for sampling until a trial is successful. Results reported for the baseline models are best for the online Deep Q-Learning model (DQN-O) which continues learning at test time. Most publications however use the slightly worse offline DQN model to compare against. Its within-template results are reasonably good (auccess scores of 77% for *Phyre-B*, 67% for *Phyre-2B*). Cross-template results are called at most reasonable by the authors, with scores of 37% for *Phyre-B* and 23% for *Phyre-2B* with DQN.

More advanced solutions for the PHYRE benchmark are proposed in (Harter et al., 2020; Li et al., 2020; Qi et al., 2020; Rajani et al., 2020; Girdhar et al., 2021; Ahmed et al., 2021; Wu et al., 2022; Li et al., 2022a; Daniel & Tamar, 2023). Of these approaches, all but Ahmed et al. (2021) investigate only the *Phyre-B* tier. Within-template, Ahmed et al. (2021) achieve the highest auccess performance at 86%. Cross-template, the highest auccess score is 56% for Li et al. (2022a), with the second best of Qi et al. (2020) at 42% quite far behind. Wu et al. (2022) do not consider the cross-template case. Ahmed et al. (2021) are the only authors to consider the *Phyre-2B* setting. They achieve 77% AUCCESS within-template and 24% cross-template here. Daniel & Tamar (2023) do not measure the benchmark metric of AUCCESS score and instead only evaluate video prediction performance of their dynamics model. Li et al. (2020) also do not report an AUCCESS number, instead they provide a plot of the success curve over number of attempts.

Overall, the *Phyre-2B* tier has not received much attention, likely because it is similar to *Phyre-B*, but considerably harder according to Bakhtin et al. (2019). Results for the *Phyre-B* tier imply that solution approaches are already capable of solving within-template tasks with very few attempts on average, while they often require 10 or more attempts cross-template.

**Contribution** PHYRE uses uncomplicated, easy-to-encode visuals consisting of a limited range of basic shapes and colors. Interaction is minimal, with only one action allowed. Arguably, the primary emphasis in PHYRE is understanding how physical objects behave over time. For this, PHYRE covers the full range of global, immediate and temporal physical variables. We see its central contribution in providing a minimalist interactive physical reasoning setting that is as simple as possible in all other aspects. This makes the benchmark an excellent tool for assessing algorithms, in an interactive setting, in terms of their forward prediction and reasoning capabilities while sidestepping the need for elaborate exploration strategies, complex input encoders or language processing.

## 2.2 Virtual Tools

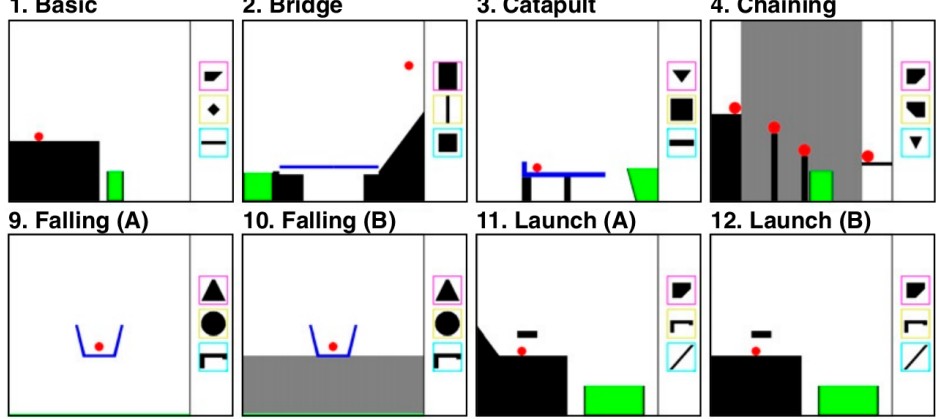

Figure 3: Twelve of the levels used in the Virtual Tools game. Players choose one of three tools (shown to the right of each level) to place in the scene to get a red object into the green goal area. Black objects (except tools) are fixed, while blue objects also move; gray regions are prohibited for tool placement. Levels denoted with A/B labels are matched pairs. Image adapted from Allen et al. (2020).

Virtual Tools (Allen et al., 2020) is a 2D puzzle (see Figure 3) where one of three objects (called tools) has to be selected and placed before rolling out the physics simulation. The goal is to select and place the tool so that a red ball ends up touching a green surface or volume. The benchmark consists of 30 levels (20 for training and 10 for testing) and embraces different physical concepts such as falling, launching, or bridging. 12 of the training levels are arranged in pairs, where one is a slight modification of the other. This allows for studying how learning is affected by perturbations of the setup.

**Solutions and related work**   As a baseline solution, the authors propose to sample random actions (sampling step) and run their ground truth simulation engine with some added noise (simulation step). The most promising simulated actions are executed as a solution with their ground truth simulation engine without noise. If this does not solve the puzzle, simulated and executed outcomes are accumulated using a Gaussian mixture model, and action probabilities are updated (update step). Sampling, simulation, and updating are iterated until the puzzle is solved. This approach is meant to model human behaviour rather than to present a pure machine learning approach similar to those used e.g. on Phyre (Section 2.1). In this capability, however, the authors find that its performance matches that of human test subjects quite well.

While the Virtual Tools benchmark has been cited as much in cognitive science research as in machine learning research, no machine learning solutions have been proposed yet. Allen et al. (2022), however, proposes a method and briefly mentions applying it on a modified version of Virtual Tools.

**Contribution**   While Virtual Tools and PHYRE (Section 2.1) share similarities, Virtual Tools puts a stronger emphasis on exploration due to the involvement of multiple tools per task. Like PHYRE, Virtual Tools serves as a valuable tool for evaluating an algorithm's physical reasoning abilities in visually simple environments. Some of the Phy-Q tasks (Section 2.3) have in common with Virtual Tools in that they require choosing tools for an action. In the case of Phy-Q, this amounts to selecting the order of birds to shoot.

## 2.3 Phy-Q

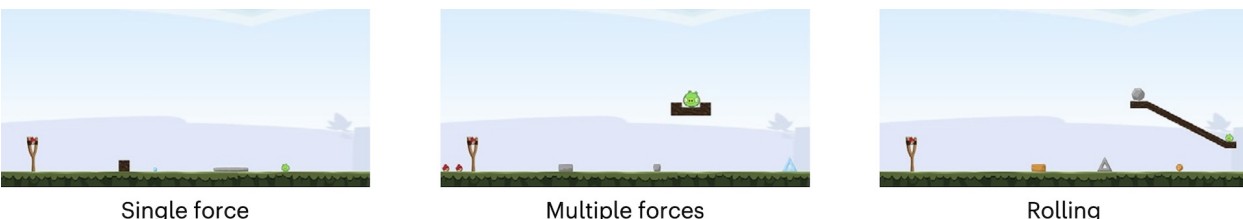

Figure 4: Four example tasks out of the 15 covered in Phy-Q. Single or multiple force means that one or several hits are required. The slingshot with birds is situated on the left of the frame. The agent's goal is to hit all the green-colored pigs by shooting birds with the slingshot. The dark-brown objects are static platforms. The objects with other colors are dynamic and subject to the physics in the environments. Image adapted from Xue et al. (2023).

The Phy-Q benchmark (Xue et al., 2023) requires the interaction of an agent with the environment to solve 75 physical Angry Birds templates that cover 15 different physical concepts (see Figure 4). Similar to PHYRE (Bakhtin et al., 2019), for each template, a set of 100 different tasks with slightly different object configurations have been generated.

There are four types of objects in the game: birds, pigs, blocks, and platforms. The agent can shoot birds from a given set with a slingshot. All objects except platforms can accrue damage when they are hit. Eventually, they get destroyed. Additionally, birds have different powers that can be activated during flight. The task in these Angry Birds scenarios is always to destroy all pigs in a scene with the provided set of birds. To achieve this goal, the agent has to provide as an action the relative release coordinates of the slingshot and the time point when to activate a bird's powers. In some scenarios, the agent has to shoot multiple birds in an order of its choosing. The observations available to the agent are screenshots of the environment and a symbolic representation of objects that contains polygons of object vertices and colormaps.

The benchmark evaluates local generalization within the same template and broad generalization across different templates of the same scenario. The authors define a Phy-Q score, inspired by human IQ scores, to evaluate the physical reasoning capability of an agent. It evaluates broad generalization across the different physical scenarios and relates it to human performance so that a score of 0 represents random, and a score of 100 indicates average human performance. In order to calculate these scores, the authors collect data on how well humans perform on their benchmarks.

**Solutions and related work**   The baseline solution uses a Deep Q-Network (DQN) (Mnih et al., 2015) agent. The reward is 1 if the task is passed and 0 otherwise. The agent can choose from 180 possible discrete actions, each corresponding to the slingshot release degree at maximum stretch. The DQN agent either learns a state representation with a convolutional neural network (CNN) or uses the a symbolic json scene description. The authors find the symbolic agents to perform notably better compared to their counterparts using convolutional frame encoders, which suggests that information retrieval from visual input is a non-negligible challenge in Phy-Q. In a modification of the baseline, a pre-trained ResNet-18 model was used for feature extraction from the first frame, followed by a multi-head attention module (MHDPA) (Zambaldi et al., 2018), followed by a DQN. Beyond their model, the authors also apply some previous agents from the AIBIRDS competition (Renz et al., 2019) to their benchmark. These, however, perform at most slightly better than their baseline. Overall, the performance of these models is significantly worse than human performance at Phy-Q scores of at most 14. There are no other available solutions as of now.

**Contribution**   Bearing similarities to PHYRE (Section 2.1) and Virtual Tools (Section 2.2), Phy-Q extends both visual fidelity, object count, and task interaction budget. As a consequence, Phy-Q can be considered genuinely more complex than the former two. Due to the requirement for ongoing interaction, exploration strategies are far more important compared to PHYRE and Virtual Tools. Within limits (since there are only few consecutive actions per environment), this qualifies Phy-Q as a comparatively simple testbed for reinforcement learning physical reasoning approaches. However, even though the visuals in Phy-Q are far from realistic, we can see the baseline agents performing better when provided with symbolic representations of the task. This implies that even though convolutional encoders are widely used in current approaches, they face challenges in effectively extracting all pertinent information from the relatively uncomplicated images that depict Phy-Q tasks. Although we did not find more complex encoder architectures (e.g. RoI Pooling layers (Girshick et al., 2014) or vision transformers (Dosovitskiy et al., 2020)) to be applied to Phy-Q, we conjecture that state of the art approaches will still be challenged w.r.t. their input encoding capabilities to some degree.

### 2.4   Physical Bongard Problems

The images in Physical Bongard Problems (PBPs) (Weitnauer et al., 2023) contain snapshots of 2D physical scenes depicted from a side perspective. The scenes in this benchmark contain arbitrary-shaped Rothgaenger et al. (2023) non-overlapping rigid objects that do not move at the time $t = t_0$ of the snapshot. There are 34 PBPs, each consisting of four scenes grouped on the left side of the image and four on the right side. The task is to predict the concept that distinguishes the scenes on the left side from those on the right side of the PBP image. Here, a concept is an explicit description that explains the difference between the scenes on the left and on the right (for instance, focusing on the stability of the objects, see Fig 5).

In general, the solution of PBPs can be based on descriptions of the whole scene or parts of the scene at any point in time or on the reaction of objects to simple kinds of interaction, e.g., pushing. This focus on indicating physical understanding by coming up with an explicit, human-readable description distinguishes the approach from more implicit and black-box-like demonstrations of understanding in the form of successful acting. The descriptions are constructed from searching a hypothesis space that encodes hypotheses as tuples [side, numbers, distances, sizes, shapes, stabilities] of a small number of hand-chosen features and object relations (such as scene side, number of its objects, inter-object distances, shapes, or stabilities). For example, the meaning of the hypothesis [left, 1-3, ?, small or large, ?, stable] is "all left scenes (and none of the right scenes) contain one to three objects that are small or large-sized and stable". The algorithm starts with all possible hypotheses and removes the incompatible ones for each scene. Finally, among the remaining

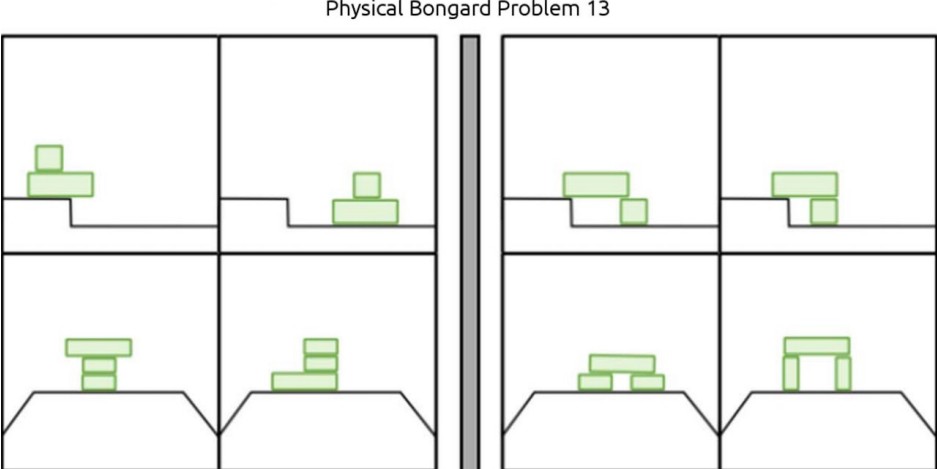

Figure 5: Example of a Physical Bongard Problem (number 13 in Weitnauer et al. (2023)). Solution: Objects form a tower vs. an arch. Image adapted from Weitnauer et al. (2023).

hypotheses, the one with the shortest length is chosen as the solution. Thus the hypothesis space can be represented as a categorization space with many possible classes.

**Solutions and related work**   Authors of PBPs provide the baseline model *Perceiving and Testing Hypotheses on Structures* (PATHS) (Weitnauer et al., 2023). There is currently no other solution model. PATHS only sees two of the scenes at a time and proceeds through a predefined sequence of scene pairs. PATHS does not take as input a pixel-based bitmap representation of scenes. Instead, each articulated object is considered a discrete entity. PATHS can perceive geometric features, spatial relations, and physical features and relations. A selector is a binary test on an object or group of objects that assesses the extent to which it exhibits a feature or some combination of features. A hypothesis consists of a selector, a quantifier, and a side. An example hypothesis is "all objects on the right side are not moving". The aim of PATHS is to construct a hypothesis that is consistent with all of the exemplars on one side of the PBP and none on the other. At each step, PATHS takes action in order to develop more perfect hypotheses. The voluntary actions are perceived, checked hypothesis, and combined hypothesis. Another type of action is to combine existing hypotheses to build more complex hypotheses. For example, "large objects" and "small objects on top of any object" can be combined into "small objects on top of large objects." The model stops as soon as a hypothesis has been checked on all scenes and is determined to be a solution, which means it matches all scenes from one side and no scene from the other side.

When comparing PATHS to human performance on PBPs, the overall success rate across different problems, the time taken to find solutions, and the impact of various presentation conditions were all taken into account. PATHS persistently searches for a solution until a predetermined maximum number of actions is reached, while humans may abandon their search or provide an incorrect answer at any point. PATHS' overall problem-solving rate is comparable to that of humans, with humans successfully solving around 45% of the problems and PATHS consistently solving around 40% of the same problems.

**Contribution**   PBP is similar to CRAFT (Section 2.5), CLEVRER (Section 2.11), and CATER (Section 2.10) in that all require agents to reason about the physical world, and they use visual stimuli to present problems to agents. Also, they use a variety of question types to assess different aspects of an agent's reasoning abilities. Although PBPs focus on physical causation and object properties, CATER focuses on action recognition and causal reasoning, CLEVRER focuses on commonsense reasoning and compositional representations of objects and scenes, and CRAFT focuses on compositional reasoning and transfer learning between different physical domains. Another property that sets PBP apart from the other benchmarks is that it does not involve physical time evolution, which is at least implicitly present in virtually all other benchmarks.

## 2.5 CRAFT

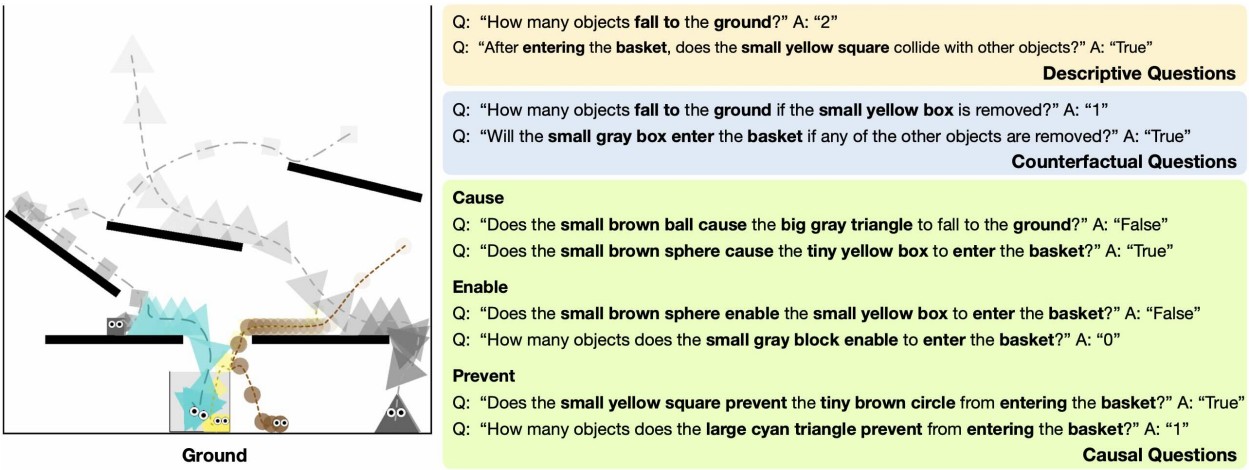

Figure 6: Example of CRAFT questions generated for a sample scene. Image from Ates et al. (2020).

The CRAFT dataset (Ates et al., 2020) is a question-answering benchmark about the physical interactions of objects presented in short videos. This dataset includes 57,000 videos and question pairs generated from 10,000 videos from 20 different two-dimensional scenes. An exemplary scene is shown in Figure 6. The figure also shows a sample of questions that are created for the scene. The types of questions in this benchmark are descriptive, counterfactual, and explicitly causal.

The representation of simulation episodes involves different data structures: video frames and the causal graph of the events in the episode that is used to generate questions. The initial and the final states of the scene refer to object properties, including the object's color, position, shape, and velocity at the start/end of the simulation. This information is provided to enable a successful answer in the benchmark.

**Solutions and related work**  The first baseline solution uses R3D (Tran et al., 2018) with a pre-trained ResNet-18 CNN base to extract information from a down-sampled video version. The text information is extracted using an LSTM. Afterward, the textual and visual information is passed to a multilayer perception network (MLP) which makes the final decision. The second baseline solution also uses R3D (Tran et al., 2018) with a pre-trained ResNet-18 CNN base to extract information from a down-sampled video version. However, instead of processing the textual and visual information separately, it processes them simultaneously using a Memory, Attention, and Composition (MAC) model (Hudson & Manning, 2018).

The baseline models are evaluated on their multiple-choice accuracy within each question category. The authors find that models perform significantly worse than humans, with 30% to 60% accuracy compared to 71% up to 87% for humans. They state that the models struggle to generalize, mostly because they fail to recognize events in videos and physical interactions between objects.

There are no available solutions, other than the baseline solutions, that cite the CRAFT paper.

**Contribution**  CRAFT is most similar to the CATER (Section 2.10) and CLEVRER benchmarks (Section 2.11) since all of these benchmarks present the agent with a video of some objects and then ask the agent questions about the physical interactions which took place. Unlike CATER and CLEVRER, CRAFT tests physical reasoning in a two-dimensional environment. One great feature of CRAFT is that is contained 57,000 videos, nearly 5 times more than CATER and CLEVRER. CRAFT contains descriptive, counterfactual, and explanatory questions, while CATER only contains descriptive questions; however, CLEVRER contains predictive questions in addition to the question types offered by CRAFT.

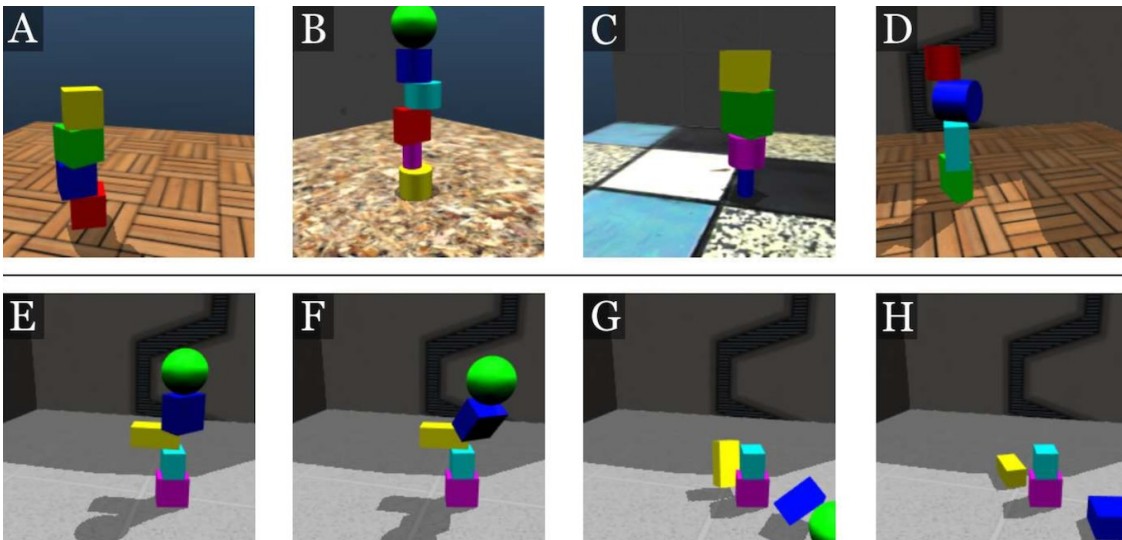

Figure 7: Different scenarios from the ShapeStacks data set. (A)-(D) initial stack setups:(A) stable, rectified tower of cubes, (B) stable tower where multiple objects counterbalance each other; some recorded images are cropped purposefully to include the difficulty of partial observability, (C) stable, but visually challenging scenario due to colors and textures, (D) violation of planar-surface principle. (E)-(H) show the simulation of an unstable, collapsing tower due to a center of mass violation. Image from Groth et al. (2018).

## 2.6 ShapeStacks

ShapeStacks (Groth et al., 2018) is a simulation-based dataset that contains 20,000 object-stacking scenarios. The diverse scenarios in this benchmark cover multiple object geometries, different complexity degrees for each structure, and various violations in the structure's stability (see Figure 7). A scenario is represented by 16 RGB images at the initial time step that shows the stacked objects from different camera angles. Every scenario carries a binary stability label and all images come with depth and segmentation maps. The segmentation maps annotate individual object instances, the object that violates the stability of the tower, the first object to fall during the collapse, and the base and top of the tower. While the actual benchmark includes per scenario only the 16 images with accompanying depth and segmentation maps, the MuJoCo world definitions are also provided to enable the complete re-simulation of the stacking scenario.

The base task in ShapeStacks is to predict the stability of a scenario, although the data additionally contains information on the type of instability, i.e. whether a tower collapses due to center of mass violations or due to non-planar surfaces.

**Solutions and related work** The two baseline solutions provided by Groth et al. (2018) use either AlexNet or Inception v4-based image discriminators along with training data augmentation to predict if a shape stack is stable. The benchmark score is computed as the percentage of correctly classified scenes from a test set that was withheld during training. The Inception v4-based discriminator performs best both on the artificial ShapeStacks scenes (Cubes only: 77.7% | Cubes, cylinders, spheres: 84.9%) as well as on real-world photographs of stacked cubes (Cubes: 74.7% | Cubes, cylinders, spheres: 66.3%).

The majority of publications that use ShapeStacks circumvent solving the stability classification problem by instead solving a related but simpler prediction problem, such as frame or object mask prediction (Ye et al., 2019; Didolkar et al., 2021; Qi et al., 2020; Ehrhardt et al., 2020; Singh et al., 2021; Engelcke et al., 2020a; 2021; Chang et al., 2022; Sauvalle & de La Fortelle, 2023a; Schmeckpeper et al., 2021; Sauvalle & de La Fortelle, 2023b; Jia et al., 2022; Emami et al., 2022). Only a few works (Fuchs et al., 2018; Engelcke et al., 2020b) directly attack solving the stability classification problem. We argue that approaches from the second group need a more refined physics understanding and can be interpreted as superior to the first group in terms of physical reasoning capabilities. Of the two approaches in this group, Fuchs et al. (2018)

only consider cube stacks but already achieve about 95% accuracy in global stability prediction by attaching small readout multilayer perceptron (MLP) models to an Inception v4 network. Engelcke et al. (2020b) obtain only 64% accuracy in stability prediction, as their model focuses on predicting viewing angle and tower height. They achieve 99% and 88% accuracy in these, respectively.

**Contribution**  The ShapeStacks benchmark covers global and immediate variables and provides diverse visual inputs comparable to real world images in complexity. CoPhy (Section 2.8), Physion (Section 2.14), and SPACE (Section 2.7) contain visually demanding stacked object scenarios as well. CoPhy focuses on more counterfactual reasoning and requires stability prediction only as one of several abilities. Physion and SPACE in addition to stacking also cover multiple physical concepts that go beyond stability prediction alone. In our assessment, the predominant challenge in ShapeStacks is extracting relevant information from the input data. While not mirroring real-world scenarios, the provided metadata alongside the stacked shape images can be quite helpful. Consequently, ShapeStacks serves as a powerful dataset and benchmark for models concentrating on information extraction from visually complex physical reasoning problems.

## 2.7  SPACE

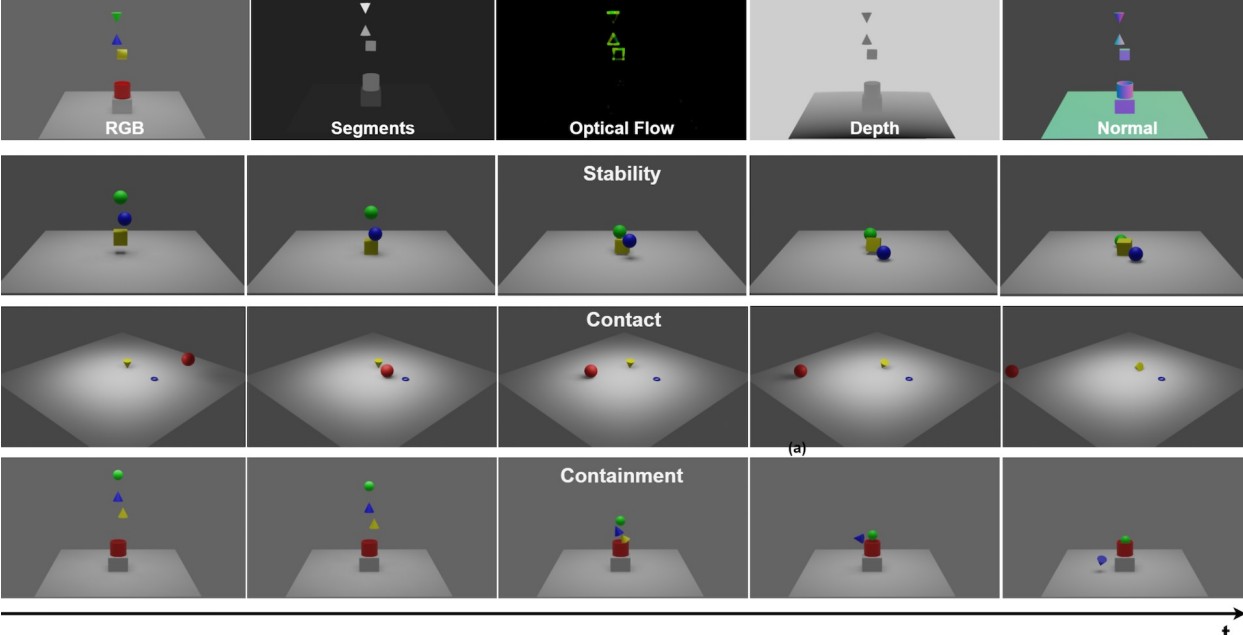

Figure 8: Example data from the SPACE benchmark. Top row: Visual data attributes for one example. The frame comprises RGB, object segmentation, optical flow, depth, and surface normal vector. Bottom three rows: Example frames from the three physical interactions. Image from Duan et al. (2021).

SPACE, introduced by Duan et al. (2021), is based on a simulator for physical interactions and causal learning in 3D environments. The simulator is used to generate the SPACE dataset, a collection of 3D-rendered synthetic videos. The dataset comprises videos depicting three types of physical events: containment, stability, and contact. It contains 15,000 (5,000 per event type) unique videos of 3-second length. Each RGB frame is accompanied by maps for depth, segmentation, optical flow, and surface normals, as shown in Figure 8.

Objects are sampled from the set $O = \{$cylinder, cone, inverted cone, cube, torus, sphere, flipped cylinder$\}$. There are three different types of videos, which come with classification labels. These imply classification tasks, while the authors use their dataset only for future frame prediction. Containment videos show a container object (colored in red and sampled from the set $C = \{$wine glass, glass, mug, pot, box$\}$) below a scene object from the set $O$. The task is to predict whether the scene object is contained in the container

object or not. Stability videos depict up to three objects from $O$ which are stacked on top of each other, and the task is to predict whether the object configuration is stable or not. Contact videos contain up to three objects at varying locations in the scene and a sphere of constant size moving around the scene on a fixed trajectory. The task is to predict whether the objects are touched by the sphere or not.

**Solutions and related work**   While, in principle, a broad range of scene classification and understanding tasks are possible due to the procedurally generated nature of the SPACE dataset, the authors do not provide all of the necessary metadata and focus on video prediction or the recognition tasks described above. They show that pretraining with the synthetic SPACE dataset enables transfer learning to improve the classification of real-world actions depicted in the UCF101 action recognition dataset. UCF101 is a large collection of short human action videos covering 101 different action categories Soomro et al. (2012). For their demonstration, they show that pretraining a model (PhyDNet, cf. below) on the SPACE dataset versus directly on the UCF101 dataset leads to improved performance when subsequently both pre-trained models are fine-tuned on the UCF101 dataset.

As of the time of writing, there are no further works that attempt to solve tasks on the SPACE datasets in the literature. The authors have, however, proposed an updated version SPACE+ (Duan et al., 2022b) of their benchmark, which quadruples the number of videos and introduces some new object classes to better evaluate model generalization.

**Contribution**   The SPACE dataset shares commonalities with ShapeStacks (Section 2.6), CoPhy (Section 2.8), and Physion (Section 2.14) in that they all involve visually taxing stacking scenarios. Similar to ShapeStacks, SPACE includes object segmentation masks and is frequently employed for tasks related to frame or object mask prediction. But in contrast to ShapeStacks, SPACE covers not only global and immediate but also temporal variables in some of its tasks. In that sense, SPACE occupies a similar position than ShapeStacks with the addition that it provides other task types beyond stacked objects. This may enable easier transfer of algorithms trained on staking scenarios to other task types, as all of them share a common look and feel.

## 2.8   CoPhy

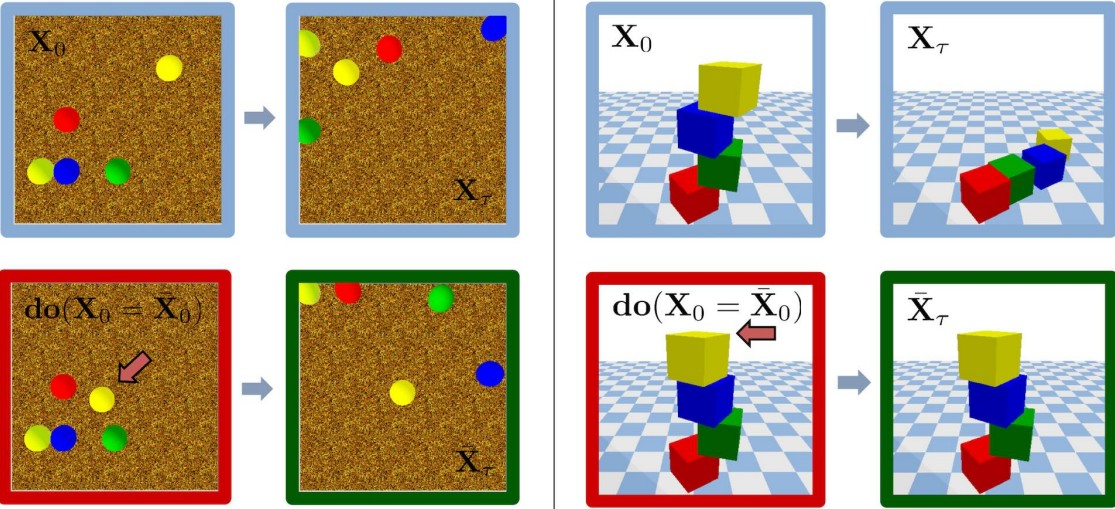

Figure 9: Exemplary CoPhy scenarios. Given an observed frame $A = X_0$ and a sequence of successor frames $B = X_{1:t}$, the question is how the outcome B would have changed if we changed $X_0$ to $\hat{X}_0$ by performing a do-intervention (e.g., changing the initial positions of objects in the scene). Image adapted from Baradel et al. (2020).

The Counterfactual Learning of Physical Dynamics (CoPhy) benchmark (Baradel et al., 2020) introduces a physical forecasting challenge that necessitates the use of counterfactual reasoning. In physical reasoning

problems, a counterfactual setting refers to a hypothetical situation where a specific aspect of the real-world scenario is changed. CoPhy tests physical understanding by evaluating what would happen if the altered condition were true. Specifically, given a video recording of objects moving in a first scenario (frames $X_{0:t}$ in Figure 9), the objective is to anticipate the potential outcome in a second scenario whose first frame $(\hat{X}_0)$ differs from the first frame in the first scenario by subtle changes in the objects' positions. The key component of this benchmark is the hidden variables associated with the objects and the environment, including mass, friction, and gravity. These hidden variables or confounders are not discernible in the single altered frame $(\hat{X}_0)$ of the second scenario, but they are observable in the video recording of the first scenario. Successfully predicting future outcomes for objects in the second scenario thus entails the estimation of the confounders from frames $X_{0:t}$ in the first scenario. For both scenarios, video recordings are provided for training of the prediction agent. The observed sequence demonstrates the evolution of the dynamic system under the influence of laws of physics (gravity, friction, etc.) from its initial state to its final state. The counterfactual frame corresponds to the initial state after a so-called do-intervention, a visually observable change introduced to the initial physical setup, such as object displacement or removal.

All images for this benchmark have been rendered into the visual space (RGB, depth, and instance segmentation) at a resolution of $448 \times 448$ px. The inputs are the initial frames (RGB, depth, segmentation) of two slightly different scenes (such as object displacement or removal), a sequence of roll-out frames of the first scene, and 3D coordinates of all objects in all available frames. The task is to predict the final 3D coordinates of all objects in the second scene. The evaluation metric is the mean squared error (MSE) of final 3D coordinates of all objects in the second scene measured between the prediction and the ground truth roll-out in a simulation.

**Solutions and related work** The proposed baseline solution model is CoPhyNet (Baradel et al., 2020), which predicts the position of each object after interacting with other blocks in the scene[2]. The inputs are RGB images of resolution $224 \times 224$. They first train an object detector to obtain the 3D location of each object. Then they use these estimates for training CoPhyNet. CoPhyNet models object interactions with a graph convolutional network (GCN), temporal dynamics with a gated recurrent unit (GRU) and predicts the final 3D coordinates of all the objects after changing the scene. The experiments show that this model performs better than MLP and humans in predicting the object's position on unseen confounder combinations and on an unseen number of blocks and balls.

Two different works have so far proposed solutions to CoPhy (Li et al., 2022b; 2020). Li et al. (2022b) extend the baseline approach by using self-attention to extract physical variables. They test their approach on the Blocktower scenario (depicted in Figure 9) and on the Collision scenario (not depicted). For Blocktower, they achieve improvements in MSE over the CoPhy baseline of up to 24% if a tower contains 3 objects, and up to 5% if a tower contains 4 objects. For the Collision scenario they achieve improvements of up to 25%. Li et al. (2020), on the other hand, omit CoPhy results in their paper even though they claim to have applied their approach to CoPhy.

Additionally, Filtered-CoPhy (Janny et al., 2022) has been proposed as a new benchmark based on CoPhy. It is a modification of the original CoPhy with a different task definition. Instead of a prediction of object coordinates, it requires future prediction directly in pixel space.

CoPhy is similar to the CRAFT (2.5), CLEVRER (2.11), ComPhy (2.12) and CRIPP-VQA (2.13) benchmarks, which also address counterfactual reasoning questions. These other benchmarks, however, do not focus as explicitly on counterfactuals and do not contain explicit interventions in their scenes. They also involve natural language tasks, which is an added complexity that is not present in CoPhy.

**Contribution** CoPhy has a unique approach to investigate counterfactual reasoning. While other benchmarks such as CRAFT (Section 2.5) provide ask counterfactual questions about videos in multiple choice, CoPhy goes beyond this by providing an explicit counterfactual initial scenario and requiring precise estimation of its final state after rollout. It is up for discussion whether this level of precision is required for every potential application, but because of this CoPhy enables a more thorough investigation of an agent's counterfactual reasoning capabilities.

---

[2]The code is available here: `https://github.com/fabienbaradel/cophy`

The second major contribution of CoPhy is its use of relational variables. These require agents to fully understand object interactions. An understanding of objects and their properties is crucial for causal reasoning, as is dictated by physics. This makes CoPhy the most rigorous of our benchmarks in testing true causal scene understanding.

## 2.9  IntPhys

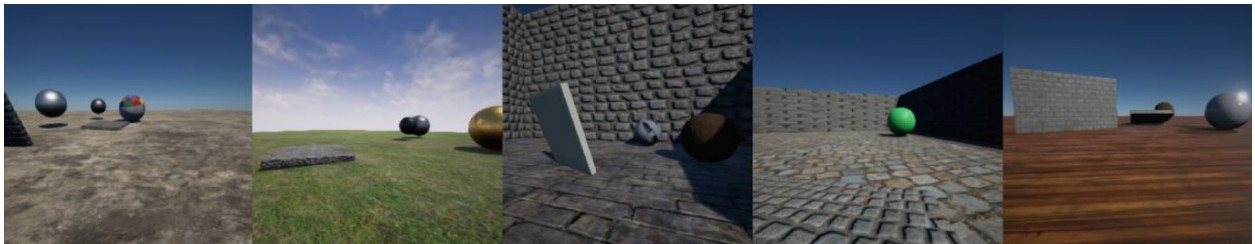

Figure 10: Exemplary screenshots from video clips used in IntPhys. Image from Riochet et al. (2021).

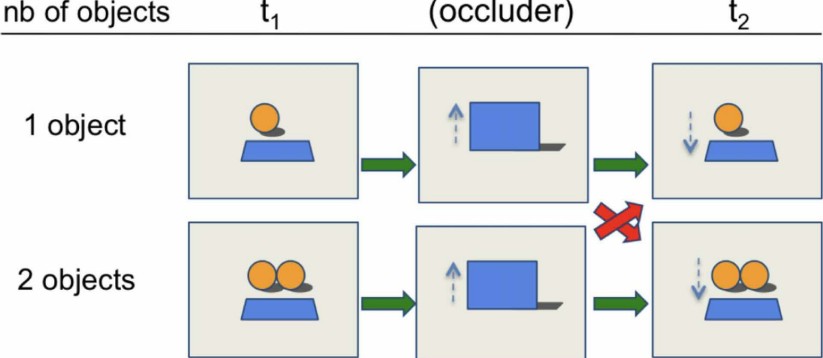

Figure 11: Illustration of the minimal sets design with object permanence in IntPhys. Schematic description of a static condition with one vs. two objects and one occluder. In the two possible movies (green arrows), the number of objects remains constant despite the occlusion. In the two impossible movies (red arrows), the number of objects changes (goes from 1 to 2 or from 2 to 1). Image from Riochet et al. (2021).

The IntPhys (Riochet et al., 2021)[3] benchmark evaluates physical reasoning for visual inputs based on the intuitive physics capabilities of infants. The benchmark is designed to measure the ability to recognize violations of three basic physical principles: Object permanence, shape consistency, and spatiotemporal continuity. Events that violate these principles (violation of expectation events – VoE) can already be detected by very young children. The benchmark consists of a set of short video clips of 7 seconds (see Figure 10) designed to present physical events that either obey the three principles (plausible scenarios) or violate at least one of them (implausible scenarios).

The model is trained on videos depicting solely plausible scenarios and subsequently evaluated on a test set that also includes implausible ones. This evaluation approach transcends the direct use of prediction error Bach et al. (2020a), as implausibility now hast to be identified indirectly by a lack of plausibility and is not a training target. If impossible videos would be part of the training set, the model may learn to distinguish them by some statistical feature unrelated to their physical plausibility. This is prevented by providing only plausible videos. The underlying principle is that if the model has successfully grasped the laws of physics, it should assign high probability to plausible scenarios, allowing low probability values to serve as indicators of implausible scenarios. Therefore, the authors claim that a model that has only been trained on plausible

---

[3]The author's original link `https://intphys.com` to this benchmark's website is outdated, please refer to Table 1 for an updated link.

scenarios should be able to generalize to other plausible scenarios but reject implausible ones. An illustration of this benchmark is shown in Figure 11.

**Solutions and related work**   The first baseline model is implemented through a ResNet that has been pre-trained on the Imagenet data and that subsequently is fine-tuned to become a classifier for the distinction of plausible versus implausible videos. One metric is the relative error rate, which computes a score within each set that requires the plausible movies to be assigned a higher plausibility score than the implausible ones. The second metric is the absolute error rate, which requires that globally, the score of plausible videos is greater than the one of implausible videos.

The second baseline model works with semantic masks of input frames and predicted future frames. The authors conclude that operating at a more abstract level is a worthwhile strategy to pursue when it comes to modeling intuitive physics.

For object permanence, the authors find that their baseline achieves an error rate of down to 0.33 if impossible events are visible and 0.5 if they are occluded. Results for shape consistency are only slightly worse with some of the baseline models. For spatiotemporal continuity, results are yet more worse with an error rate of down to 0.4 for visible and 0.5 for occluded events. Error rates in human test subjects, in comparison, are reported to be 0.18 and 0.3 for object permanence, 0.22 and 0.3 for shape consistency, and 0.28 and 0.47 for spatiotemporal continuity. The first of these numbers refers to visible impossible events and the second to occluded events.

While the IntPhys benchmark has been widely cited and regularly discussed, there are only two proposed solutions in the literature by Smith et al. (2019) and Nguyen et al. (2020). The latter do not differentiate in their report between visible and occluded impossible events. They merely report error rates of 0.24, 0.17 and 0.28, respectively, for object permanence, shape consistency and spatiotemporal continuity. Smith et al. (2019) only test their approach on the object permanence task. They, too, do not differentiate between visible and occluded impossible events and report an error rate of 0.27 for this task. Beyond these approaches, Du et al. (2020) propose an object recognition model and suggest using it for physical plausibility downstream tasks.

Overall, we conclude that there is not much research into the IntPhys task. The work that exists so far does either not report results thoroughly or performs considerably below the human baseline. The IntPhys website (see link in Table 1) contains a leaderboard of approaches, although it is not clear what approaches have been used for the different submissions.

Beyond IntPhys, there are two additional VoE benchmarks that involve physical reasoning, which are, however, very similar to IntPhys both in their covered concepts and their data format (Piloto et al., 2022; Dasgupta et al., 2021).

**Contribution**   IntPhys was chronologically the first physical reasoning benchmark of the ones considered in this paper. It is inspired by neuroscience, similar to Virtual Tools (Section 2.2), while most other benchmarks feature tasks inspired by machine learning capabilities or applications. As a consequence, the psychologically inspired reasoning task of judging physical plausability is what sets IntPhys apart from other benchmarks.

To test performance on this task in a way similar to how humans would be tested, IntPhys seeks to reproduce realistic scenes with high-fidelity visuals. While this makes the task of reasoning more difficult and introduces the additional challenge of dealing with rich background information, it also means that agents successful on this benchmark are more likely to perform well in realistic environments.

## 2.10   CATER

CATER (Girdhar & Ramanan, 2019) is a spatiotemporal reasoning benchmark that extends CLEVR (Johnson et al., 2017), which is based on static images. It provides blender-based rendering scripts to generate videos and associated classification tasks such as action recognition and target tracking. The videos contain simulated 3D objects that move on a 2D plane, as shown in Figure 12. For each video, objects have been sampled randomly from a small set of generic bodies (such as cubes, cones, or cylinders) and come with a set

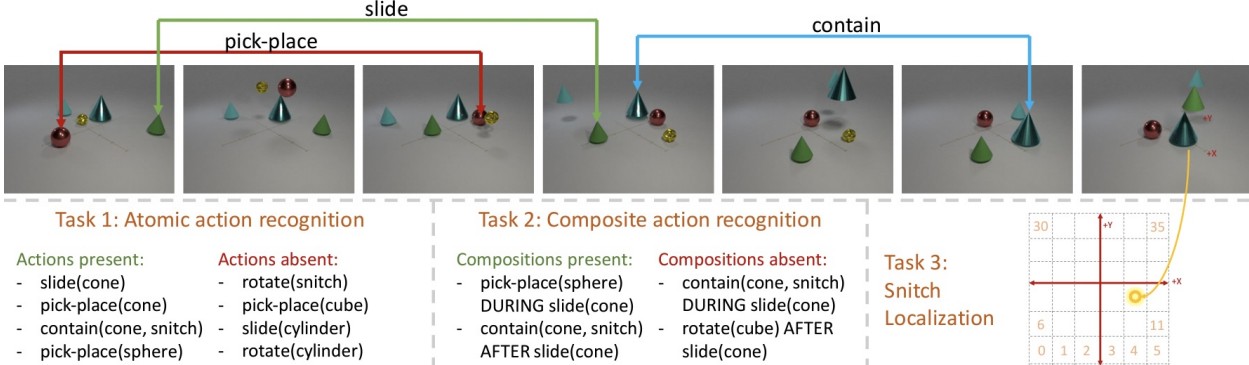

Figure 12: CATER dataset and tasks. Sampled frames from a random video from CATER. Some of the actions afforded by objects in the video are labeled on the top using arrows. Three tasks are considered: Task 1 requires a listing of all actions that are observable in the video. Task 2 requires identifying all observable compositions (i.e. temporal orders) of actions. Task 3 requires quantified spatial localization of a specific object (called snitch) that became covered by one or more cone(s) and, therefore, has disappeared from direct view while still being dragged with the movements of the cone(s) enclosing it. This task tests agents for the high-level spatial-temporal understanding that is required in order to track an invisible object through the movements of its occluding container. Image from Girdhar & Ramanan (2019).

of permitted atomic actions such as sliding and placing. These actions are assigned and applied randomly to each object throughout a video. The authors provide a pre-rendered set of 16500 videos that were created in this manner as their CATER dataset on which they perform their baseline experiments.

The benchmark comes with three predefined classification tasks: The first task, atomic action recognition, requires detecting all pairings between atomic actions and objects that have occurred within a video. The second task, compositional action recognition, requires the detection of temporal compositions (only temporal pairs are considered) of object-action pairings, together with their temporal relation (before, during or after). The last and final task, snitch localization, is only applicable to a subset of videos. It requires locating a certain object, called a snitch object, that has become covered and dragged around by one or multiple nested cones (the only object type with this container property). The answer needs to specify the location only as a discrete grid cell to allow a classifier approach to this last task.

**Solutions and related work** The authors stress that they extend existing benchmarks beyond two frontiers: The tasks include detecting temporal relationships between actions, and the videos avoid scene or context bias, i.e., there is very little information about the task solution present in the background of video frames. Both aspects encourage solution approaches capable of temporal reasoning rather than analyzing individual frames. As baseline solutions, a few existing methods are adapted and compared. The base method used for all three tasks employs temporal CNNs (Wang et al., 2016; 2018), which process individual frames or short frame sequences. Results are aggregated by averaging over all frames of a video. As an alternative aggregation approach, the authors also employ an LSTM. In addition, they apply an object tracking method (Zhu et al., 2018) to locate the occluded snitch object. While the models work reasonably well for the simplest task of identifying atomic actions, performance breaks down to rather mediocre accuracy for compositional action recognition. Snitch localization, finally, does not work well with any of the methods. In general, LSTM aggregation yields higher accuracy scores than aggregation by averages.

Various papers have addressed the tasks posed by CATER. Solutions for tasks one and two (atomic and compositional action recognition) have been proposed in Samel et al. (2022), Kim et al. (2022), and Singh et al. (2022a). Solutions to the third task requiring hidden object localization have been proposed in Goyal et al. (2021); Ding et al. (2021a); Traub et al. (2022); Zhou et al. (2021); Zhang (2022); Harley et al. (2021); Faulkner & Zoran (2022); Castrejon et al. (2021); Sun et al. (2022); Luo et al. (2022).

For task one, Singh et al. (2022a) report a 2% improvement in mean average precision (mAP) over baseline, at a value of 96%. For task two, they report a 17% improvement over baseline at 80% mAP. Samel et al. (2022) and Kim et al. (2022) only report results for task two. Samel et al. (2022) achieve a performance of 69% mAP. They do however show that their model performs better for out-of-distribution cases than comparative models. Kim et al. (2022) do not propose a full new architecture, but rather a different sampling strategy for video data. Using this, they were able to improve the performance of other models on CATER's second task by 3% to 10%. For task three, many of the proposed solutions compare their results amongst each other. A very recent paper with a very thorough reporting of results of various solution approaches is Traub et al. (2022) and we refer the reader to their table for a closer look. Overall, top 1 and top 5 accuracies reached on the object localization task exceed 90% and 98%, respectively. These results exceed those of the baselines by far.

Beyond approaches to solve the proposed tasks, CATER has been used for object segmentation tasks in Kipf et al. (2021); Singh et al. (2022b); Bao et al. (2022); Frey et al. (2023) and unsupervised physical variable extraction in Kabra et al. (2021). Furthermore, some authors have created new datasets based on CATER, for object tracking tasks (see Van Hoorick et al. (2022); Shamsian et al. (2020)), for video generation from compositional action graphs (see Bar et al. (2020)), for video generation based on images and text descriptions (see Hu et al. (2022); Xu et al. (2023)), and for dialogue systems concerning physical events (see Le et al. (2021; 2022)). All these additional datasets are approached by the respective authors specifically for particular tasks, but they nonetheless touch on various aspects of physical reasoning as well.

**Contribution** CATER covers a diverse range of global, immediate and temporal physical variables in its tasks. By asking questions specifically about object-action relations, the benchmark directly encourages solution approaches that have object-level task understanding. Furthermore, the temporal aspect of the benchmark suggests approaches that feature a form of internal model which understands physical causality. While generally both aspects have shown to be advantageous for the majority of benchmarks, learning them remains mostly indirect as they are merely a tool to generate a better answer to some question. In contrast to that, CATER provides inputs and training targets specifically tailored to learn about those two aspects of physical reasoning.

## 2.11 CLEVRER

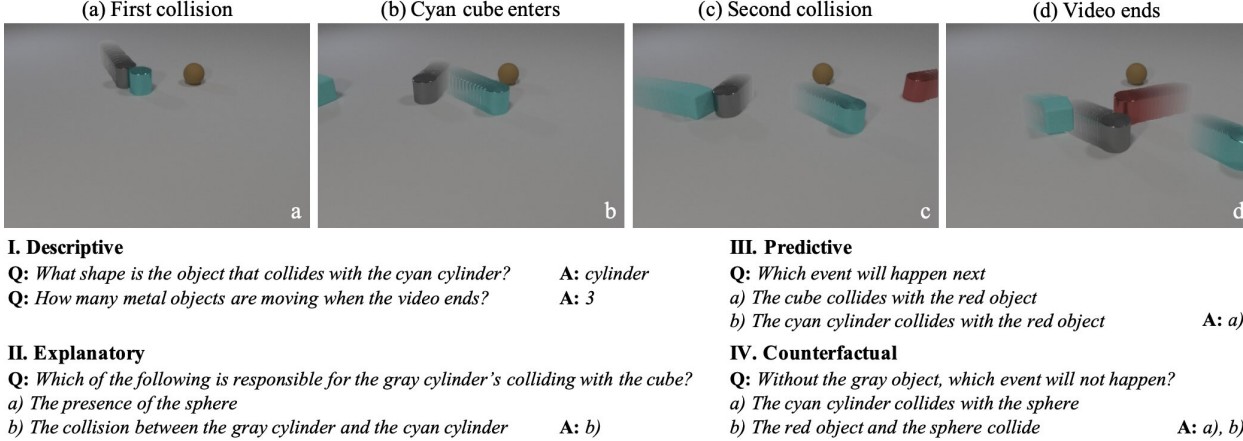

**I. Descriptive**
**Q:** *What shape is the object that collides with the cyan cylinder?*    **A:** *cylinder*
**Q:** *How many metal objects are moving when the video ends?*    **A:** *3*

**II. Explanatory**
**Q:** *Which of the following is responsible for the gray cylinder's colliding with the cube?*
*a) The presence of the sphere*
*b) The collision between the gray cylinder and the cyan cylinder*    **A:** *b)*

**III. Predictive**
**Q:** *Which event will happen next*
*a) The cube collides with the red object*
*b) The cyan cylinder collides with the red object*    **A:** *a)*

**IV. Counterfactual**
**Q:** *Without the gray object, which event will not happen?*
*a) The cyan cylinder collides with the sphere*
*b) The red object and the sphere collide*    **A:** *a), b)*

Figure 13: Examples of videos and questions in CLEVRER. They are designed to evaluate whether computational models can answer descriptive questions (I) about a video, explain the cause of events (II, explanatory), predict what will happen in the future (III, predictive), and imagine counterfactual scenarios (IV, counterfactual). In the four images (a–d), the blurred motion traces are only provided to reveal object motion to the human observer; they are absent in the input to the recognition system. Image from Yi et al. (2020).

Collision Events for Video Representation and Reasoning (CLEVRER) (Yi et al., 2020) is a benchmark that requires temporal and causal natural language question answering. Like CATER above, it extends CLEVR

(Johnson et al., 2017) from images to short, artificially generated 5-second videos along with descriptive, predictive, explanatory, and counterfactual questions about their contents. Each video contains multiple instances of cubes, spheres, or cylinders that can have one of eight different colors and consist of either a shiny or a dull material. The combination of object geometry, material, and color is unique within a video, and all objects are placed on a white, planar surface. The objects are either sliding across the surface, resting, or colliding with each other (see Figure 13), with the videos generated such that often multiple cascading collisions occur.

CLEVRER contains 10,000 training videos, 5,000 validation videos, and 5,000 test videos. They are accompanied by a total of 219,918 descriptive (D), 33,811 explanatory (E), 14,298 predictive (P), and 37,253 counterfactual (C) questions, which were generated procedurally together with the videos. The training input consists of a video and a text question. Models treat each question as a multi-class classification problem over all possible answers. For example, in the case of descriptive questions, answers can be given in natural language, and in this case, the correct answer is a single word, which can be specified with a model that outputs a softmax distribution over its vocabulary. Descriptive questions are evaluated by comparing the answered token to the ground truth, and the percentage of correct answers is reported. Multiple choice questions are evaluated based on two metrics: The per-option accuracy, which measures the correctness of the model regarding a single option across all questions, and the per-question accuracy, which measures the percentage of questions where all choices were correct.

In the three remaining categories, tasks are essentially binary classification problems. Answering the questions correctly is taken as an indicator for properly identifying, understanding, and predicting the dynamics of the video.

**Solutions and related work**   The first baseline solution uses a convolutional LSTM to encode the video into a feature vector and then combines this feature vector with the input question using a MAC network (Shi et al., 2015) to obtain a final prediction. The second baseline solution encodes the video information using a convolutional neural network (CNN) and encodes the input question using the average of the question's pre-trained word embeddings (Mikolov et al., 2013). These two embeddings are then passed to an MLP or LSTM to make the final prediction. While these baselines do not generally learn object-centric representations, the authors also try to combine them with object-centric representations and achieve superior performance. Accuracies for these baseline models reach up to 86% (D), 22% (E), 42% (P) and 25% (C).

In addition to their baselines, the authors of CLEVRER also propose what they call a neuro-symbolic oracle model. It consists of a Mask R-CNN (He et al., 2017) to process video frames, a seq2seq model (Bahdanau et al., 2014) to parse questions, and a propagation network (Li et al., 2019) as a dynamics model. The first two models are trained individually. This oracle model achieves accuracies of 88% (D), 79% (E), 68% (P) and 42% (C).

Solution approaches for CLEVRER include (Chen et al., 2021; Ding et al., 2021b; Sautory et al., 2021; Ding et al., 2021a; Zhao et al., 2022; Wu et al., 2022). Of these, the Aloe model by Ding et al. (2021a) has long been seen as state of the art, with accuracies of 94% (D), 96% (E), 87% (P) and 75% (C). As such it outperforms most of the cited approaches, but is surpassed for predictive questions by Wu et al. (2022) who reach 93% accuracy, and for counterfactual questions by Zhao et al. (2022), who reach 78% accuracy. Additionally, Zablotskaia et al. (2021) propose a model for scene decomposition which they test on CLEVRER videos without considering CLEVRER's questions.

**Contribution**   With its four categories of questions CLEVRER has defined useful classes of basic reasoning tasks, and with it founded a new subfield of physical reasoning. The baseline performances prove that these different tasks cover a range of reasoning challenges that differ in complexity as well as difficulty. Various question types have since been adapted by CRAFT (Section 2.5), ComPhy (Section 2.12), and CRIPP-VQA (Section 2.13). Providing tasks in natural language requires successful reasoning approaches to become closer to practical usefulness in real-world applications featuring human-machine interaction. Beyond its questions, CLEVRER's visually clear videos help to investigate reasoning capabilities of approaches without results being confounded by scene decomposition. Due to this advantage, similar videos have been adapted in CATER (Section 2.10), SPACE (Section 2.7) and CRIPP-VQA.

## 2.12 ComPhy

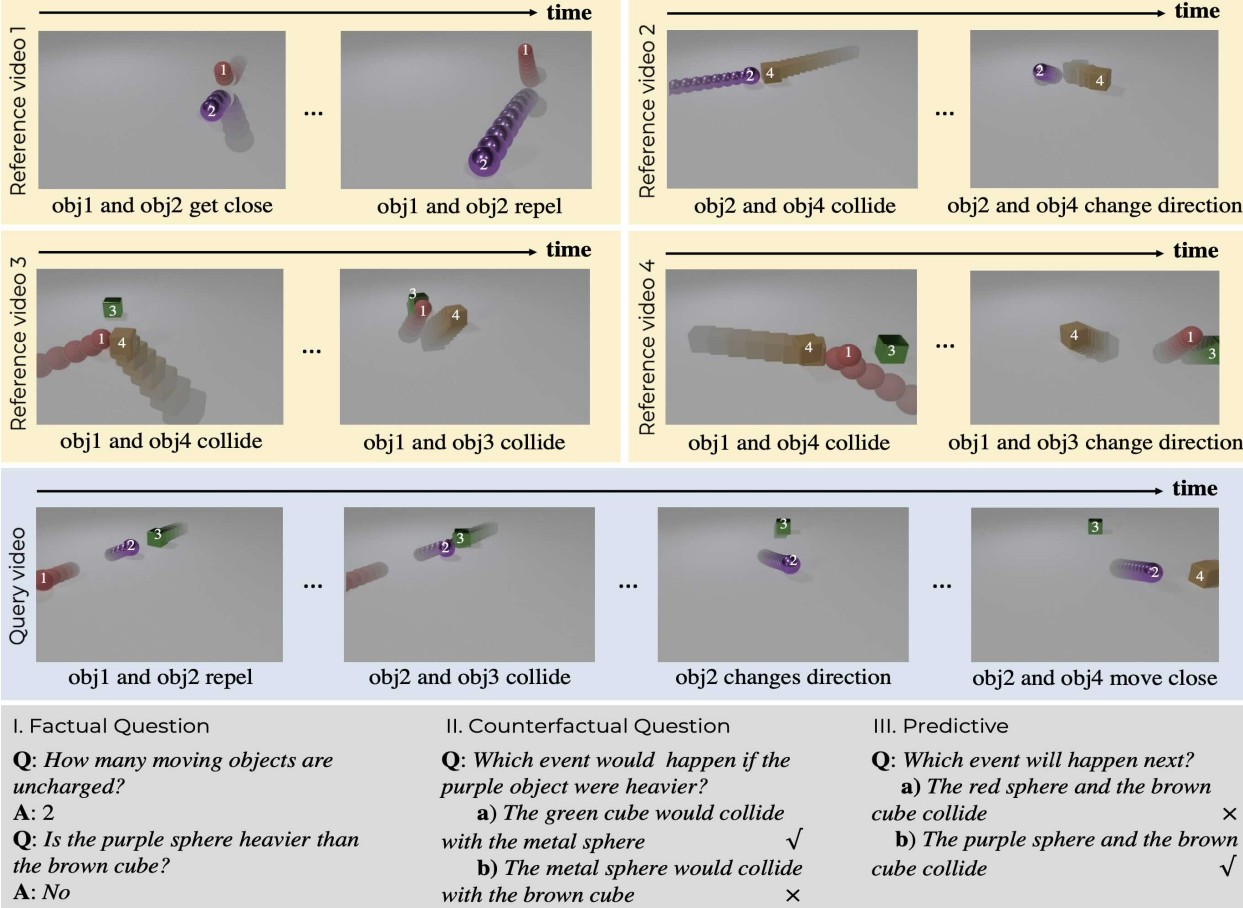

Figure 14: Sample target video, reference videos, and question-answer pairs from ComPhy. Image from Chen et al. (2022).

ComPhy (Chen et al., 2022), shown in Figure 14, is a visual question answering benchmark for reasoning about hidden physical properties such as the charge and mass of objects. The authors speak of intrinsic as opposed to extrinsic properties, although we chose to call them relational as opposed to immediate variables in this survey (see also Section 1). In contrast to immediate, visible variables, relational variables are invisible and only become apparent through interaction between objects. This means relational variables are only observable over a time interval rather than in any particular instant.

The benchmark consists of videos that show the temporal evolution of up to 5 simulated 3-dimensional objects that interact while moving on a plane. The objects have immediate properties of color, shape, and material and discrete relational properties of mass (light or heavy) and charge (negative, neutral, positive). Interaction between objects can take the form of attraction/repulsion or collision.

The videos come in atomic sets of 5, divided into 4 reference videos and one target video for few-shot learning. Reference videos are 2 seconds long and show object interaction but don't contain labels for physical properties, while the target video is 5 seconds long and comes with ground truth on physical properties (both relational and immediate variables), and only contains objects seen in at least one reference video. The actual train, validation, and test sets of the benchmark contain a few thousand atomic video sets, and while the physical properties of objects are consistent within atomic sets, they are assigned randomly between different atomic sets.

The task in ComPhy is to answer natural language questions either about facts in the target video, predictions about the evolution of the target video, or counterfactual questions concerning alternate outcomes if the relational properties of objects had been different. Factual questions are open-ended, while predictive and counterfactual questions need to be answered in multiple-choice, where several options can be simultaneously correct. This effectively amounts to multi-label classification.

**Solutions and related work**  As baseline solutions, the authors test adaptations of the models CNN-LSTM (Antol et al., 2015), HCRN (Le et al., 2020), MAC (Hudson & Manning, 2018) and ALOE (Ding et al., 2021a). They also provide a small study on human performance. Additionally, they provide their own solution approach called compositional physics learner. It consists of several modules which perform perception, property inference, dynamic prediction, and symbolic reasoning. This model achieves better performance than the baseline approaches at 80%, 56% and 29% per-question accuracy for factual, predictive and counterfactual questions respectively, but worse than the human testers who achieved 90%, 75%, 52%, respectively..

The only work that has proposed an approach to address the ComPhy benchmark, as of now, is by Tang et al. (2023). They compare their method not to the compositional physics learner, because it is trained in a supervised way. Compared to the unsupervised baseline models, however, they achieve a slight improvement in predictive and counterfactual questions at 33% and 25%, respectively. In factual questions, they perform worse than baseline at 59%.

**Contribution**  ComPhy is a benchmark that takes question types and video setups from CLEVRER (Section 2.11), but extends these with relational variables. These are its main contribution. The only other benchmark with true relational variables is CoPhy (Section 2.8), although CoPhy contains different relational variables and is not centered around the difficulties they provide. As discussed in Section 2.8, relational variables are crucial for causal reasoning but very underrepresented in physical reasoning benchmarks (see Table 2). This makes ComPhy an important benchmark. It is best suited to evaluate an agent's ability to extract relational variables and it is the only one which features natural language questions concerning in this context.

## 2.13  CRIPP-VQA

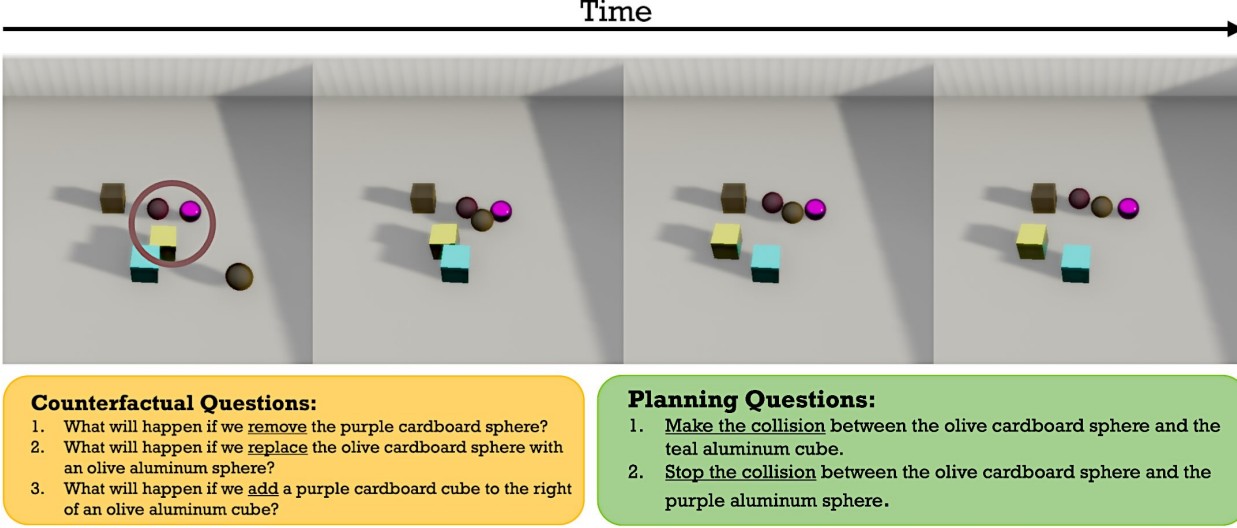

Figure 15: Stills from a video in CRIPP-VQA with exemplary counterfactual and planning questions. Image adapted from Patel et al. (2022).

The Counterfactual Reasoning about Implicit Physical Properties Video Question Answering benchmark, in short CRIPP-VQA (Patel et al., 2022), extends the field of classical video question answering into the domain of physical reasoning. It focuses on relational physical variables, specifically mass and friction.

The benchmark consists of 5,000 videos, split into train, validation, and test sets, plus an additional 2000 videos to evaluate out-of-distribution generalization, where objects might have previously unseen physical properties. Each video is 5 seconds long and shows a 3D scene of computer-generated objects (cubes and spheres) moving on a plane. Stills are shown in Figure 15. One object is initially in motion, or two in the out-of-distribution videos. This sets off a series of collisions, which are affected by the friction and mass of each object.

Friction and mass of objects are assigned to unique combinations of immediate variables {Shape, Color, Texture}. This means mass and friction are important to extract but are not truly relational, as they can be deduced from individual frames once this mapping is known. This makes them effectively immediate. In the out-of-distribution videos, however, this assignment is removed, and mass and friction become truly relational.

The videos are accompanied by about 100,000 natural language questions, each assigned to an individual video. Of these questions, 45% are counterfactual, 44% descriptive, and 11% concern planning. Descriptive questions ask facts about the scene, for instance, about a count, material, or color. Planning questions ask which action would have been necessary to obtain a certain different final state. The possible actions are to add, remove, or replace objects. Both descriptive and planning questions require language answer tokens. The counterfactual questions ask how the final scene would have differed if the initial condition had been different, and multiple natural language answers are provided together with the question. The task is to predict for each of the provided answers whether it is true or false, which means the counterfactual questions are essentially multilabel classification tasks.

For all categories and videos, however, success during evaluation is measured by accuracy in the answers given to questions. Either accuracy per option, in the counterfactual case, or accuracy per question.

**Solutions and related work**   As baseline solutions, the authors test the MAC (Hudson & Manning, 2018) compositional VQA model, the HRCN VQA model (Le et al., 2020), and the Aloe visual reasoning model (Ding et al., 2021a), modified to work on Mask-RCNN features (He et al., 2017) and using BERT-generated word embeddings (Devlin et al., 2018). The modified Aloe model generally performs best out of these three baselines at 71% accuracy for descriptive questions, 63% accuracy for counterfactual questions and 32% accuracy for planning questions. In the out-of-distribution case, the authors state that performance drops to close to random. In addition to these baseline models, the authors present results achieved by a small group of humans ($n = 6$) for comparison, who achieve 90%, 78% and 58% accuracy on descriptive, counterfactual and planning questions, respectively.

As of the time of writing, there are no further approaches proposed for CRIPP-VQA in the literature.

**Contribution**   CRIPP-VQA is similar to CLEVRER (Section 2.11) and ComPhy (Section 2.12) since these depict similar 3D collision scenes on a plane and also involve natural language questions. However, as opposed to ComPhy, the part of CRIPP-VQA that is not out-of-distribution does not have truly relational variables, and the question types covered in CRIPP-VQA do only partly overlap with both ComPhy and CLEVRER.

Beyond counterfactual and descriptive questions, which are also featured in other benchmarks, CRIPP-VQA is the only benchmark to include planning questions. This makes it unique and introduces a whole new reasoning task to the four introduced by CLEVRER (Section 2.11). The results of both the baselines and the human test subjects suggest that planning is a task that is considerably more difficult than even counterfactual questions, which were the hardest task in CLEVRER. The reason for this is likely that beyond causal reasoning, planning requires an understanding of consequences of actions. In this regard, CRIPP-VQA is related to the interactive benchmarks of this survey while being the only one that does not feature an actual simulator to train the agent on. The use of natural language makes tasks more realistic but introduces further complexity.

## 2.14 Physion

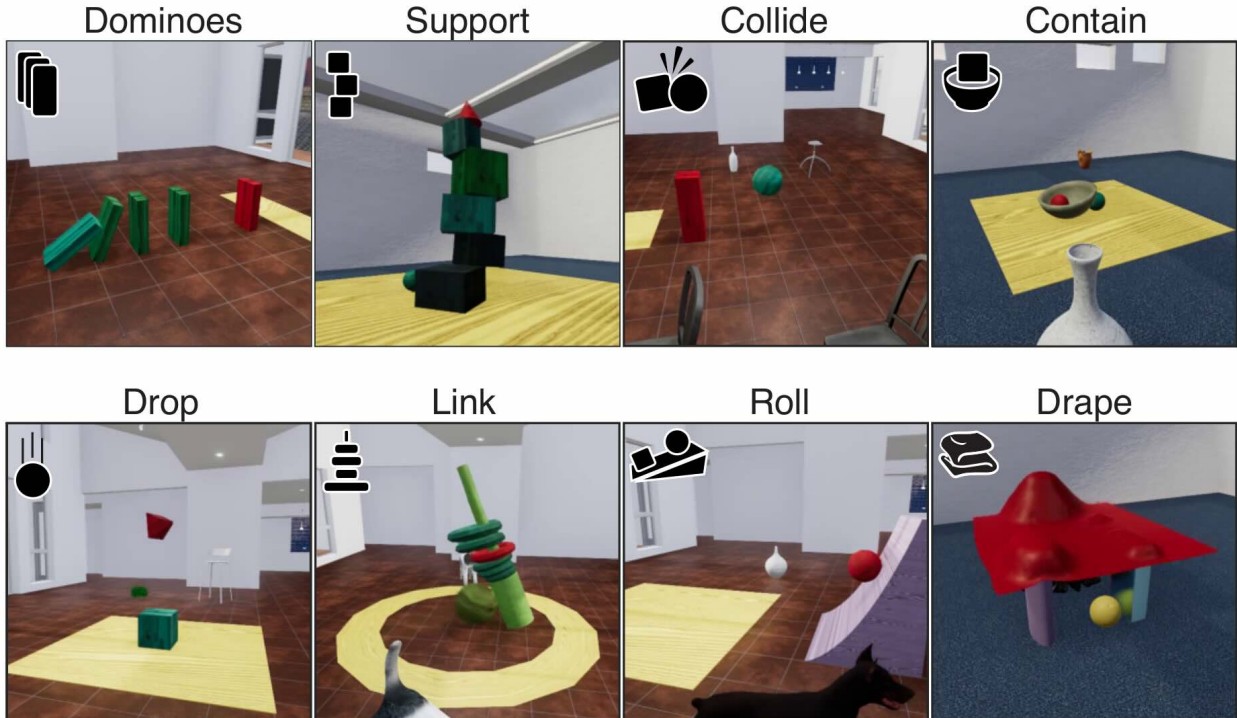

(a) Stills from the eight different physical concepts in the Physion benchmark.

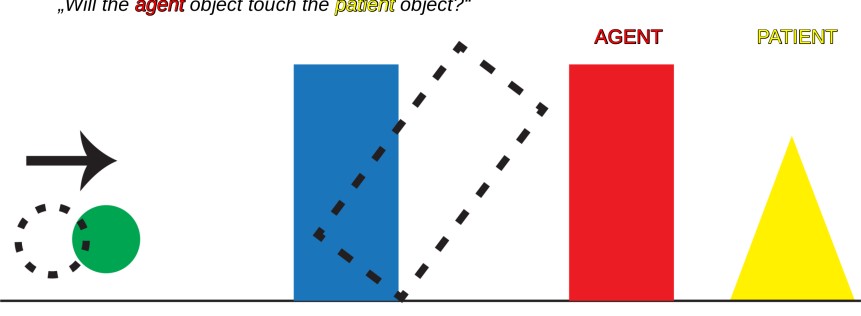

(b) Simplified explanation of the Physion task.

Figure 16: Stills and task explanation of the Physion benchmark. The videos contain an agent (red) and patient (yellow), and possibly a probe (green) that moves and hence initiates the unfolding of the scene. Images are adapted from Bear et al. (2021).

Physion (Bear et al., 2021) contains a set of realistic 3D videos that are 5-10 seconds long and present 8 different physical interaction scenarios: dominoes, support, collide, contain, drop, link, roll, and drape (see Figure 16a). The benchmark is evaluated on an object contact prediction (OCP) task, which asks whether two selected objects, called *agent* and *patient* (see Figure 16b) will touch throughout the video. A video ends after all objects come to rest.

The videos, also called stimuli, are rendered at 30 frames per second. The following data is supplied. 1.) visual data per frame: color image, depth map, surface normal vector map, object segmentation mask, and optical flow map; 2.) physical state data per frame: object centroids, poses, velocities, surface meshes (which can be converted to particles), and the locations and normal vectors for object-object or object-environment collisions; 3.) stimulus-level labels and metadata: the model names, scales, and colors of each object; the

intrinsic and extrinsic camera matrices; segmentation masks for the agent and patient object and object contact indicators; the times and vectors of any externally applied forces; and scenario-specific parameters, such as the number of blocks in a tower. All stimuli from all eight scenarios share a common OCP task structure. There is always one object designated the *agent*, and one object designated the *patient*, and most scenes have a *probe* object whose initial motion sets off a chain of physical events. Machine learning models and human test subjects are asked to predict whether the agent and patient object will have come into contact before or until the time all objects come to rest.

The Physion dataset consists of three different parts: *dynamics*, *readout*, and *test* sets. The *dynamics* training set contains full videos without *agent* and *patient* annotations or labels that indicate the outcome of the OCP task. It is meant to train representation or dynamics models from scratch. The *readout* training set presents only the first 1.5 seconds of videos and the corresponding OCP task labels to separately train the model for solving the OCP tasks with a frozen pre-trained dynamics model. Lastly, the *test* set contains only the initial 1.5 seconds of videos and OCP task labels and is meant to evaluate trained models.

**Solutions and related work**   While Physion is meant to benchmark machine learning models, the authors evaluate it on human volunteers to establish a baseline. These volunteers achieve an accuracy of between 65% and 90% for the various physical concepts. Against this baseline, they compare a range of models which can be categorized into CNN-based vision models and graph-based dynamics models. Some of the vision models learn object-centric representations, while the graph-based dynamics models require a preexisting object-graph representation instead of raw visual input. Their experiments show that vision models perform worse than humans, although those with object-centric representations generally perform better (about 55% to 65%) than those without (about 45% to 55%). The graph-based dynamics models can sometimes compete with humans, with the best model performing only a few percentage points worse than human test subjects. This leads to the authors concluding that the main bottleneck is learning object-centric representations from visual scenes.

Approaches to tackle the Physion benchmark have been proposed in Wu et al. (2022); Han et al. (2022); Nayebi et al. (2023). Wu et al. (2022) report an accuracy of 67% across all concepts, and additionally report that the RPIN method by Qi et al. (2020) obtains 63% accuracy. Han et al. (2022) report an accuracy of 60% to 89% across different concepts. Nayebi et al. (2023), finally, focus on sensory-cognitive modeling and focus their reporting on the correlation of the performance of different models with the performance of human test subjects. Beyond these approaches, Physion videos are also used explicitly for the task of video prediction in Nayebi et al. (2023); Lu et al. (2023).

Physion++ is proposed in Tung et al. (2023) as a second, newer version of Physion. Its videos are based on the same physics engine as Physion, but the benchmark goes beyond Physion in that it focuses on relational variables and shows more object interactions in its videos. Models trained on Physion++ videos are thus expected to explicitly infer relational variables in order to solve its tasks.

**Contribution**   With its high-fidelity videos and different physical scenarios, Physion puts its focus on the physics rather than the reasoning. The reasoning task is a classification of the final state after rollout of the simulation, which is similar to many other benchmarks. The physical scenarios in Physion, however, are often more complex than in other benchmarks. In particular the link and drape concepts (see Figure 16a) do not appear in any other benchmarks. This makes Physion challenging on one hand, but, of those we cover here, also the most realistic in terms of real-world physics and inputs.

Additionally, Physion stands out by the models that have been tested on it by the authors. The paper as well as the website (see Table 1) contain useful comparisons of not only different models but also different model classes. This gives rise to multiple insights, such as that object-centric representations enable agents to perform consistently better than unstructured latent representations.

## 2.15 Language-Table

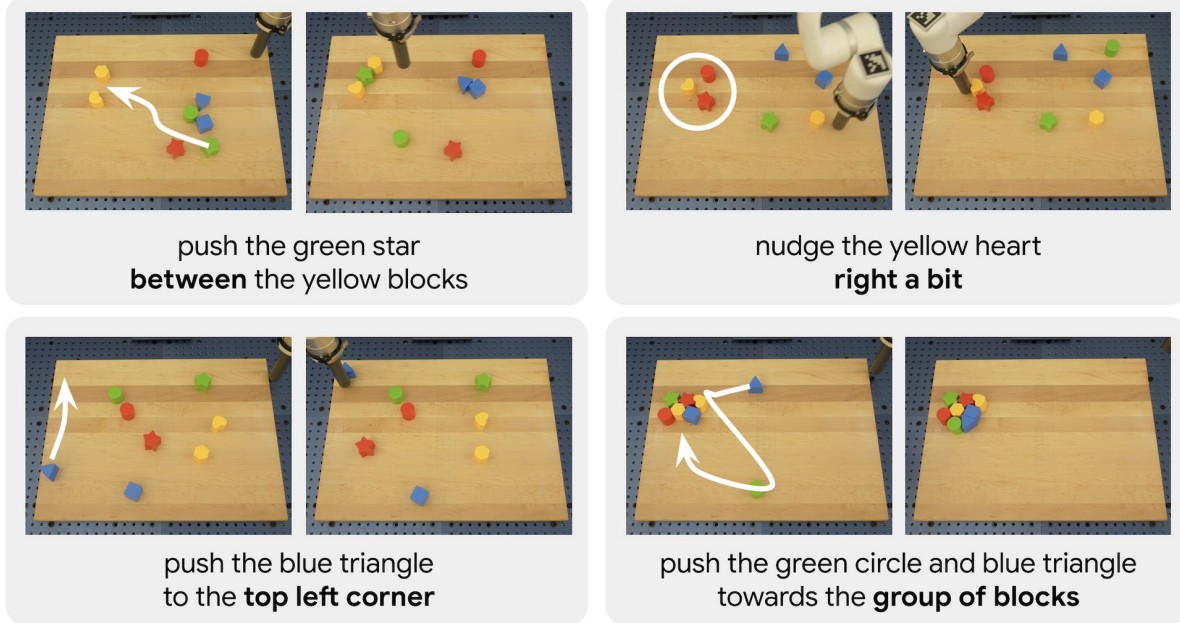

Figure 17: Language-Table rollouts on a sample of the more than 87,000 crowdsourced natural language instructions with a wide variety of short-horizon open vocabulary behaviors. Image from Lynch et al. (2022).

The Language-Table dataset (Lynch et al., 2022) contains nearly 600,000 natural language-labeled trajectories of a robotic arm moving blocks placed on a table according to language instructions (see Figure 17). An example instruction is: "Slide the yellow pentagon to the left side of the green star." This benchmark is intended for imitation learning for natural language-guided robotic manipulation. Each trajectory in the dataset records the state of a UFACTORY xArm6 robot with 6 joints and a video of the state of the table. The table is made of smooth wood and has 8 plastic blocks (4 colors and 6 shapes) placed on it. The dataset contains 413,000 trajectories gathered from real-world data and 181,000 simulated trajectories. The average episode length for real-world data is 9.9 minutes $\pm$ 5.6 seconds. The average episode length for simulated data is 36.8 seconds $\pm$ 15 seconds

Additionally, the authors release the *Language-Table environment*, which is a simulated environment closely matching the real-world setup used in the dataset. This environment is useful for evaluating potential solutions and hyper-parameter tuning. Using the Language-Table environment they also create the Language-Table benchmark. This benchmark computes automated metrics for 5 task families with a total of 696 unique task variations. The task families are as follows: block2block, block2abs, block2rel, block2blockrel, and separate. block2block tasks require the agent to push one block to another. block2abs tasks require the agent to push a block to an absolute location on the board (e.g. top-left, bottom-right, center, etc.). block2rel tasks require the agent to push a block in a relative offset direction (e.g. left, down, up, and right, etc.), block2blockrel tasks require the agent to push a block such that it is offset from the target block in a particular direction (left side, top right side, ... of block X). Separate tasks require the agent to separate two blocks. For all tasks, success is a binary variable that is true if the distance between the source block and the target location/block is below a valid threshold.

**Solutions and related work**   To solve the Language-Table benchmark, the authors propose a multi-stage process. They begin by using a pre-trained ResNet model (He et al., 2016) to extract visual features from the current input video frame and use the CLIP (Radford et al., 2021) to embed the natural language instruction into a visual latent space. Next, they train a language-attends-to-vision transformer (Vaswani et al., 2017) whose keys and values are based on the ResNet embeddings, and queries are based on the CLIP

embeddings. The output of this transformer is considered to contain the current frame's state. Afterward, the last $n$ results from the language-attends-to-vision transformer gathered from the last $n$ frames are then given to a temporal transformer whose output is given to a ResNet MLP, which will predict the current frame's action.

With this method, the authors achieve a 93% success rate for short-horizon instructions, and a 85% success rate for long-horizon instructions when agents get real-time language feedback. Without it, their performance on long-horizon instructions drops to 25% success rate. A method to solve the Language-Table tasks has been proposed by Driess et al. (2023). Their results vary from 30% to 70% success rate for different tasks if the model has seen only 10 demonstrations, up to 58% to 90% when the model has seen 40 or 80 demonstrations. These results are all for long-horizon instructions. Furthermore, Xiao et al. (2022); Rana et al. (2023); Yang et al. (2023) have used the Language-Table dataset for training their models, although they have solved different tasks such as robotic behavior prediction.

**Contribution** While the other language-related benchmarks require the agent to answer physical reasoning questions, Language-Table's unique feature is that it tests an agent's ability to follow physical reasoning-related natural language instructions. Additionally, it is very accurate to real life, allowing practitioners to test in more realistic scenarios. Finally, Language-Table contains nearly 500,000 trajectories, which is a lot more than other benchmarks.

## 2.16 OPEn

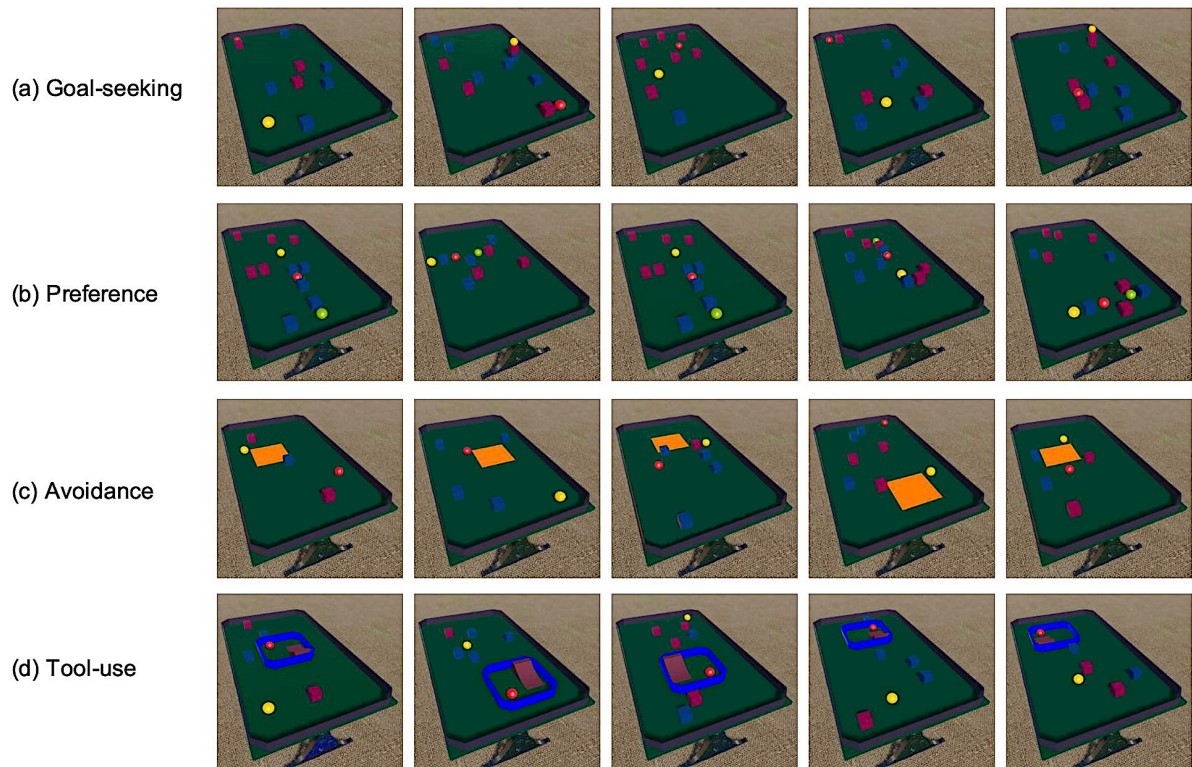

Figure 18: OPEn evaluation suite for 4 downstream physical reasoning tasks. Image from Gan et al. (2021).

OPEn, introduced by Gan et al. (2021)[4], is a 3D interactive, open-ended physics environment (see Figure 18). It is comprised of a table framed by raised borders, with objects placed on top of it. Interactions happen through a rolling agent that observes visual data and can move on the table. The agent can execute actions to

---

[4]The author's original link http://open.csail.mit.edu to this benchmark's website is outdated, please refer to Table 1 for a link to the GitHub repository instead.

move in one of eight directions for a fixed distance and interact with the objects in the scene by colliding with them. OPEn consists of two modes. The first, called sandbox, provides randomly generated non-episodic environments, which are used to learn state representations or even a physics model by interactive exploration without a specific task. The second, the evaluation suite, expresses tasks through reward functions of a reinforcement learning environment. The tasks are to move towards a goal, move towards a preferred object, avoid a region on the table, use a ramp to escape a region of the table, and seek a goal. The tasks are meant to assess models trained on the sandbox.

**Solutions and related work**  The authors test four baseline solutions on their task, i.e., they first run a sandbox and then the evaluation suite phase. The tested approaches are a vanilla PPO agent (Schulman et al., 2017), an intrinsic curiosity module (Pathak et al., 2017), random network distillation (Burda et al., 2018), and CURL (Laskin et al., 2020). Each of the latter three is also tested with RIDE rewards (Raileanu & Rocktäschel, 2020). The best-performing solution is CURL with RIDE rewards. However, the authors note that none of the baselines benefit meaningfully from the sandbox pretraining mode, which suggests that the methods used are unable to build rich, general world models in the absence of downstream tasks. The proposed baselines thus do not comprehensively solve the OPEn benchmark. Beyond these baselines, there are no proposed solutions to this benchmark as of the time of publication.

**Contribution**  OPEn is the only benchmark in this collection which specifically considers itself a reinforcement learning problem. This continuous interaction, as opposed to the one-time interaction of PHYRE (Section 2.1) and Virtual Tools (Section 2.2) has implications further discussed in Section 3.3. The line between reinforcement learning problems Bach et al. (2020b) and physical reasoning problems is fuzzy. OPEn does not only specifically consider itself a benchmark rather than just an environment, it also has a focus on physical reasoning in a similar way to Language-Table (Section 2.15). Similar to multiple other benchmarks, OPEn puts a focus on the interaction of objects and tools, which are crucial to understand in order to achieve a goal.

## 2.17   Other Benchmarks

This section includes benchmarks which test physical reasoning ability but are not directly focused on the topic. Strong performance on these tasks requires agents to possess a significant number of other skills alongside physical reasoning. As such, we feel it may be difficult for practitioners to assess the physical reasoning abilities of their agents using solely these tasks. However, that being said, these benchmarks are still valuable under the right circumstances.

**Transformation Driven Visual Reasoning benchmark (TRANCE)**  This benchmark (Hong et al., 2021) considers two images, one for an initial state and the other for a final state. The aim is to infer the transformation of the objects between the two images. Transformation refers to any kind of change in the object. The benchmark evaluates how well different machines can understand the transformations. TRANCE is created based on the CLEVR dataset (Johnson et al., 2017), which depicts objects that are characterized by five types of attributes, including shape, size, color, material, and position; TRANCE adopts the same default values for the attributes of its objects as CLEVR. To start generating the benchmark starts with randomly sampling a scene graph, similar to the initial step of CLEVR. In the second step, the questions and the answers are getting generated with a functional program based on the scene graph (the first step).

**Watch & Move**  This is a benchmark (Netanyahu et al., 2022) about one-shot imitation learning. An agent watches an expert rearrange 2D shapes and is then confronted with the same shapes in a different configuration. The agent needs to understand the goal of the expert demonstration and arrange its own shapes in a similar way. The authors propose to use graph-based equivalence mapping and inverse RL to achieve this.

Table 3: Input and target formats for all benchmarks. The *Input Data* column lists the kind and format of data a solution approach can make use of. The *Target Data* column lists training targets. For interactive benchmarks, the target data usually consists of actions that are input for a simulator and thus differ from the final performance measure. For non-interactive benchmarks, the target data equals the final performance measure or is easily mappable to it.

| | Benchmark | Input Data | Target Data |
|---|---|---|---|
| 2.1 | PHYRE | **Images:** 256x256px 7-channel (each channel is distinct color) | 3D or 6D action vectors for positions and radii of red ball(s) |
| 2.2 | Virtual Tools | **Images:** 600x600px RGB **Metadata:** Used tool, tool position, number of solution attempts | Integer representing the tool and 2D position vector |
| 2.3 | Phy-Q | **Images:** 480x640px RGB **Metadata:** Symbolic task representation in JSON format | Either integer $n \in \{0, ..., 179\}$ representing slingshot angle or 3D vector representing x/y coordinates of pulled back the slingshot and the activation time of birds with special abilities |
| 2.4 | Physical Bongard Problems | **Images:** 102x102px RGB | Class label |
| 2.5 | CRAFT | **Images:** 256x256px RGB **Questions:** Natural language text | Class label |
| 2.6 | ShapeStacks | **Images:** 224x224px RGB, depth, object segmentation | Stability flag |
| 2.7 | SPACE | **Images:** 224x224px RGB, depth, object segmentation, surface normal, optical flow | **Video prediction:** RGB image of next frame **Scenarios:** True/false flag |
| 2.8 | CoPhy | **Images:** 448x448px RGB, depth, object segmentation | 3D coordinates of objects |
| 2.9 | IntPhys | **Images:** 228x228px RGB, depth, object segmentation **Metadata:** Object positions, camera position, object IDs | Plausibility score between 0 and 1 |
| 2.10 | CATER | **Images:** 320x240px RGB | Class label |
| 2.11 | CLEVRER | **Images:** 480x320px RGB **Questions:** Natural language text | **Descriptive questions:** Answer token **Other:** Class label |
| 2.12 | ComPhy | **Images:** 480x320px RGB **Questions:** Natural language text | **Factual questions:** Answer tokens **Other:** Class labels |
| 2.13 | CRIPP-VQA | **Images:** 512x512px RGB **Metadata:** Object locations, object velocities, object orientations, collision events **Questions:** Natural language text | **Counterfactual questions:** Boolean flags **Other:** Answer tokens |
| 2.14 | Physion | **Images:** 512x512px RGB, depth, object segmentation, surface normal, optical flow **Metadata:** Physical properties of objects, collision events, force vectors, names of scene objects, segmentation masks, number of total objects, etc. | Flag that indicates whether agent and target object touch or not |
| 2.15 | Language-Table | **Images:** 320x180px RGB **Instructions:** 512D token vector **Metadata:** (x,y) of gripper and (x,y) of target positions | 2D action vector to move gripper |
| 2.16 | OPEn | **Images:** 168x168px RGB (can be configured to other resolutions) | Integer $n \in \{0, ..., 8\}$ representing the action label |

Table 4: Benchmarks assigned to the groups introduced in Section 3. Each group relates to a capability required by agents to solve this benchmark.

| Interactive 3.1 | Concept recognition 3.2 | World model 3.3 | Language-related 3.4 |
|---|---|---|---|
| 2.1 Phyre | 2.4 Physical Bongard Problems | 2.1 Phyre | 2.5 CRAFT |
| 2.2 Virtual Tools | 2.5 CRAFT | 2.2 Virtual Tools | 2.11 CLEVRER |
| 2.3 Phy-Q | 2.6 ShapeStacks | 2.3 Phy-Q | 2.12 ComPhy |
| 2.15 Language-Table | 2.7 SPACE | 2.5 CRAFT | 2.13 CRIPP-VQA |
| 2.16 OPEn | 2.9 IntPhys | 2.8 CoPhy | 2.15 Language-Table |
| | 2.10 CATER | 2.13 CRIPP-VQA | |
| | 2.12 ComPhy | 2.14 Physion | |
| | 2.14 Physion | 2.15 Language-Table | |
| | | 2.16 OPEn | |

## 3 Benchmark Groups

The ultimate goal in physical reasoning AI is to develop generalist, powerful physical reasoning agents. In this section, we group benchmarks based on four reasoning capabilities that we consider critical parts of a future generalist agent. Currently, almost all agents aim to solve individual benchmarks, and most benchmarks focus on a particular capability. We believe that, eventually, a fully capable physical reasoning agent should be able to perform well across all groups of benchmarks.

The four core capabilities are the following: A general agent should (*i*) be interactive, i.e. able to explore the world by acting in it instead of just passively observing. It should also (*ii*) be able to recognize known physical concepts and categories. It should further (*iii*) be able to build models of world dynamics to a point where it can extrapolate into the future and make predictions. Finally, agents should (*iv*) be capable of language in order to reason and communicate on abstract yet semantically meaningful concepts rather than merely images, numbers, or categories.

Below, we discuss the implications of each capability in detail and group the benchmarks according to which capabilities they predominantly cover. The benchmarks assigned to individual groups are also listed in Table 4.

### 3.1 Interactive Benchmarks

From human and animal studies, we know that active exploration of the world is an important prerequisite for building a solid understanding of the laws governing it. In that sense, interactive benchmarks provide the unique option of active hypothesis testing. The agent can – and has to – decide what to try next and how to fill the gaps in its world knowledge.

Interactive physical reasoning tasks are often formulated as reinforcement learning (RL) problem since it caters to the sequential nature of physical processes, and there is extensive prior work on balancing exploration and exploitation as well as dealing with uncertainty. We found that RL approaches are used in solution algorithms for all interactive benchmarks in our review.

Due to their interactive nature, interactive benchmarks usually have to rely on an internal simulator that dynamically reacts to inputs to generate data on the fly. It is up to the agent to collect useful and unbiased data, while non-interactive benchmarks may come with carefully balanced datasets. Since it is infeasible to test all possible input-output combinations of the simulator beforehand, it has to be as accurate and numerically stable as possible to achieve good extrapolation beyond what could be tested during development.[5]

While all interactive benchmarks (see Table 1) listed in this survey focus on achieving a specific final state of the simulation, other tasks such as question answering or counterfactual reasoning are conceivable and illustrate yet untapped potential of interactive physical reasoning tasks. To sum up, we see interactivity as

---

[5]An example of why inaccurate simulators are problematic are physics glitches that have been seen exploited by solution algorithms to circumvent or greatly simplify the proposed task.

an important requirement on the way to generalist physical agents and deem it to define one of the relevant groups for semi-generalist agent development.

## 3.2 Concept Recognition Benchmarks

In physical reasoning, it is a difficult problem to learn explicit, basic physical concepts with a model that has no prior knowledge of physics. Agents, especially future generalist agents, however, benefit from explicit knowledge of physical concepts relevant to their reasoning task. Various current benchmarks pose classification tasks which effectively teach such concepts. By training on a physically meaningful classification target, an agent will learn to recognize the concept.

In the case of IntPhys (Section 2.9), for instance, the physical concept and classification target is physical plausibility, which requires object permanence, shape constancy, and spatio-temporal continuity. The PBP benchmark (Section 2.4), on the other hand, requires the identification of a concept that defines the difference between two physical scenes. In the case of CATER (Section 2.10), the classification target is the temporal order of actions.

Physical scenarios can be decomposed into concepts on various levels of abstraction and granularity, as shown by these examples, which makes this group rather broad. Beyond IntPhys, PBPs, and CATER, it also contains CRAFT, ShapeStacks, SPACE, ComPhy, and Physion. What they have in common, however, is that they require models to learn to recognize physical concepts. The remaining two benchmarks, PHYRE and Virtual Tools involve classification tasks. These two are interactive benchmarks where the classification target concerns reaching a desirable state in the environment rather than recognizing a physical concept.

In current benchmarks, classification or concept recognition happens for its own sake and does not form part of more complex reasoning chains. In the future, however, this group might well extend to benchmarks that do not explicitly train models in a supervised manner on given concepts. Instead, benchmarks might expect models to learn concepts in an unsupervised way and as a part of more complex reasoning tasks. To evaluate concept understanding, they can then check whether concepts were explicitly learned during an evaluation phase.

## 3.3 World Model Benchmarks

World model benchmarks can be seen as a group of tasks that test the capability of an agent to make use of an internal model of the task behavior. In passive benchmarks, the model can be harnessed to simulate future task dynamics and in interactive benchmarks, to predict consequences of different ways of acting in order to identify and compare feasible solution paths and their relative pros and cons.

World models vary with respect to what they predict, e.g. forward models predict outcomes caused by inputs, and inverse models predict inputs required to cause desired outputs. They also vary w.r.t. their domain and ranges (which can either be narrow or wide, discrete or continuous). Often, the modeled relationship is probabilistic, e.g., represented as a joint probability density of inputs and outputs. Depending on the entropy of this density, the model may provide very weak or very distinct predictions. This includes model-free approaches, where an agent starts with a tabula rasa model that is shaped into a more or less predictive world model as a result of (supervised or unsupervised) learning, and on the other hand, approaches that start with a very detailed world model that predicts deterministic and unique outcomes that are highly accurate within its domain of applicability. Models can also take other forms, such as, e.g. attention mechanisms or value functions in reinforcement learning. Finally, adaptive models can also include predictions about the confidence (or confidence intervals) with (or within) which their predictions are valid.

Although a world model can be expected to boost performance in most physical reasoning tasks, it may be hard to obtain, and the cost of obtaining and using it always has to be weighed off against its usefulness. For instance, tasks that contain high levels of branching (e.g. due to uncertainty or stochasticity) are usually not good candidates for world models as the model predictions are likely to quickly become unreliable. On the other hand, if the task dynamics exhibit only little or no branching at all, a world model is a promising tool to increase performance. One concrete example of this is by Ahmed et al. (2021), who demonstrate that

adding the predictions of a learned dynamics model to their PHYRE solution algorithm notably increases performance compared to a variant without the dynamics model.

An ultimate generalist physical reasoning agent should be able to combine a set of world models, which complement each other w.r.t. different forms of reasoning, along with further factors, such as range, precision, reliability, and computational effort in order to cope with the huge range of different physical contexts that need to be covered in real-world situations. Moreover, the models with a clear focus on physics need to be aided by models that cover regularities beyond physics, such as the mind states of other agents and how these are affected by factors that are, in turn, physical. This suggests architectures that arrange models in suitable hierarchies to avoid solving exhaustive and expensive search problems, as can be seen, e.g. in Ahmed et al. (2021); Li et al. (2022a); Qi et al. (2020); Rajani et al. (2020). To avoid irrelevant simulation candidates in the first place, intuitive physics understanding could be leveraged as a guide for the precise world model so that the latter is used more economically. Whether being guided carefully or used for exhaustive searches, we believe precise world models are an important and powerful tool in the arsenal of a physical reasoning agent. Since a large portion of solution approaches across all benchmarks we list in this survey uses some form of world model, membership in this group is fairly fuzzy. We argue that predominantly benchmarks which either strictly require a world model or obviously profit from it due to being interactive, should be members of this group. Thus, we see Phyre (2.1), Virtual Tools (2.2), Phy-Q (2.3), CRAFT (2.5), CoPhy (2.8), CRIPP-VQA (2.13), Physion (2.14), Language Table (2.15) and OPEn (2.16) as core members of this group. For benchmarks with higher visual fidelity and/or more interacting objects, learning a world model can be harder in practice and thus may be less likely to increase performance, although most benchmarks have various solution approaches that use world models.

## 3.4  Language-related benchmarks

Benchmarks that comprise natural language processing or generation extend physical reasoning problems to the domain of natural language processing. These benchmarks require agents to harness the descriptive power of language to assess, describe, and/or solve physical problems. In contrast to images, language can represent different aspects of a physical process in varying levels of abstraction and detail, potentially putting the focus on certain aspects while neglecting others. However, this comes at the cost of a higher potential for ambiguity, redundancy, synonymy, and variation. Another advantage of using natural language is an easier comparison of state-of-the-art agents to human physics understanding. On the other hand, adding language understanding and maybe even generation to the stack of problems makes natural language-based physical reasoning benchmarks a considerably harder problem class in general. Nonetheless, we see natural language physical reasoning benchmarks as an important group to master on the way to a generalist agent.

Table 5: Comparison of all benchmarks which require the agent to answer natural language questions about a given visual input (i.e., video or images). The output format can be *multi-label P(all_Tokens | Q)* for a multi-label probability distribution over all possible answers conditioned on the current question and input, allowing for multiple correct answers, or *single-label P(all_Tokens | Q)* for a single-label probability distribution permitting only one correct answer. Mere *multi-label* means a multi-label probability distribution over only a small set of possible answers conditioned on the current question and input. Evaluation criteria *multi-CA* and *single-CA* mean multi-label or multi-class (but single-label) classification accuracy, respectively.

| Benchmark | Visual Input | Question Format | Output Format | Evaluation Criteria |
|---|---|---|---|---|
| 2.5   CRAFT | video | raw text | single-label P(all_Token\|Q) | single-CA |
| 2.11  CLEVRER | video | raw text | multi-label P(all_Token\|Q) | single-CA, multi-CA |
| 2.12  ComPhy | video | raw text | multi-label, single-label P(all_Token\|Q) | multi-CA, single-CA |
| 2.13  CRIPP-VQA | video | raw text | multi-label P(all_Token\|Q) | single-CA, multi-CA |

The passive language-related benchmarks of Section 2 require the capacity of question answering. Table 5 lists these benchmarks and their properties. Of the passive benchmarks, we consider CRAFT (2.5),

CLEVRER (2.11), and CRIPP-VQA (2.13) as the core members because they are the only benchmarks that require agents to answer arbitrary questions provided via raw text. This requires agents to possess language understanding ability to solve these benchmarks. The remaining passive benchmarks do not involve a text modality, but can still be transformed into question answering tasks in order to evaluate the physical reasoning capacity of language-based models.

Additionally, there is one interactive benchmark, Language-Table (2.15), which does not require question answering but rather the following of language instructions. It is also a core member of the category of language-related benchmarks.

## 4 Physical Reasoning Approaches

This section presents a brief and general introduction to solution approaches for physical reasoning tasks. Among the multitude of solution approaches for the presented benchmarks, each approach includes a variety of inductive biases, carefully crafted loss functions and often multiple modules that have been tuned to work together. A comprehensive survey of physical reasoning approaches is therefore beyond the scope of this work and we encourage the reader to make themselves familiar with solution approaches based on the lists of references we provide with each benchmark.

For further reading on the topic of physical reasoning approaches, Duan et al. (2022a) provide a survey of approaches for modelling intuitive physics. However, their tasks and solution approaches overlap only partially with the ones we consider here based on the list of benchmarks. Beyond the physical reasoning community, approaches are additionally inspired by methods from other fields such as visual reasoning (Małkiński & Mańdziuk, 2023), visual question answering (Zhong et al., 2022), and embodied AI (Duan et al., 2022c). To solve a physical reasoning task, solution approaches need to extract task-relevant information from the provided input (i.e. *perceive* the task), relate and process the information (i.e. *reason* about the task), and synthesize an answer that solves the proposed task. While monolithic approaches, such as Li et al. (2022a), can perform physical reasoning with a single, large neural network, the majority of solution approaches reflect these three steps via a modular design principle that combines individual components into a larger algorithm. This preference for modular approaches likely stems from their ability to provide greater control over the reasoning process and to yield deeper insights into their functionality. The encoder connects the physical reasoning algorithm to the outside world and generates a latent representation given arbitrary input modalities. If multiple input modalities are available, different encoder architectures may be combined. Besides simple multilayer perceptrons (MLPs) and convolutional neural networks (CNNs), approaches often use more advanced encoder components:

- Pretrained large CNNs, (e.g. Zhang (2022); Kim et al. (2022))

- Region of interest pooling layers (Girshick et al., 2014)

- Transformers (e.g. Dosovitskiy et al. (2020))

- (Convolutional) Recurrent neural networks (RNNs) (e.g. Keren & Schuller (2017))

Although Zhang (2022); Kim et al. (2022) succeed by using unstructured latent representations in their models, object-centric encoders (Locatello et al., 2020; Kipf et al., 2021; Engelcke et al., 2021; Daniel & Tamar, 2022; Singh et al., 2022b), which are structured and trained to represent e.g. positions or velocities of individual objects, are preferred in most approaches. In line with that, Yi et al. (2020) and Bear et al. (2021) find that models with object-based representations consistently outperform those with unstructured latents on the CLEVRER (2.11) and Physion (2.14) benchmarks, respectively.

Object-centric encoders that only consider individual images in a video usually require methods to track objects across frames in order to obtain consistent object identities. This is achieved either classically with a Hungarian algorithm (e.g. in Nguyen et al. (2020); Zhou et al. (2021)) or enforced directly through the model architecture (e.g. in Kipf et al. (2021), Daniel & Tamar (2023)).

The reasoning component processes latent representations and, depending on the task, may be explicitly implemented as a forward dynamics model. Although not universal, the vast majority of benchmarks explicitly feature forward dynamics models as part of their reasoning components. While simple dense MLPs can be used, the following more advanced dynamics models are common:

- RNNs (e.g. Traub et al. (2022); Chen et al. (2022); Bear et al. (2021); Baradel et al. (2020))

- GNNs (e.g. Ye et al. (2019); Li et al. (2022b); Han et al. (2022))

- Transformers (e.g. Wu et al. (2022); Sun et al. (2022); Qi et al. (2020); Goyal et al. (2021))

RNNs tend to be used more in earlier papers and baseline solutions proposed with benchmarks. Transformers, on the other hand seem more popular nowadays and are usually part of dedicated reasoning approaches rather than benchmark baselines. GNNs, finally, have been popular over the years due to their strong inductive bias for modeling interactions. While transformers and RNNs can in principle operate on any representation, GNNs require a graph structure that contains meaningful disentangled object representations.

The answer synthesizing component operates on the latent representations produced by the reasoning component and can be considered one of two output decoder types. Depending on the nature of the reasoning task, it predicts latent object states, physical variables, images, or natural language. To achieve this, a spectrum of architectures is harnessed, including MLPs, CNNs, RNNs, and transformers, with a discernible preference emerging for the latter. While a second type of decoder that recreates inputs from latent representations is often added to provide a clear training signal for the rest of the model, its application for solving the actual physical reasoning tasks remains rare. Simple CNN decoders are a common choice in case of image data, but there are also more complex approaches like transformers or RNNs.

An important difference between passive and interactive benchmarks is that in case of the latter, the space of possible answers is commonly larger and the feedback for answers might be delayed, which complicates the learning process. To address these new challenges, we found solutions to leverage reinforcement learning principles and resort to actor and/or critic networks that either operate on the output of the encoder, the reasoning component, or both. The technical implementation of actor and critic networks is usually done via simple MLPs.

## 5 Discussion

In this work, we have compiled a comprehensive collection of physical reasoning benchmarks that encompass various input and output formats across different task domains. Each benchmark is accompanied by a detailed description, including information about input-output formats, task nature, similarities to other benchmarks, and solution approaches. None of the physical reasoning benchmarks we have encountered so far encompasses all types of physical reasoning, and none of them pose challenges in all available task dimensions. We propose the utilization of groups to address these challenges, employing semi-generalist agents that can serve as a stepping stone toward the development of a truly generalist physical reasoning agent.

While the space of possible solutions to physical reasoning tasks is potentially vast, we nonetheless have found a clear structure emerging from the solution approaches for the presented benchmarks. Competitive approaches rely exclusively on deep neural networks to tackle the benchmarks. Although a small minority of papers apply monolithic models to the input data to directly solve a physical reasoning task, most rely on intricately structured neural network architectures that contain encoders, forward models, decoders and task solution models. Approaches for interactive benchmarks can add reinforcement learning actor and/or critic networks. Hereby, each of the listed components may be represented by complex MLPs, CNNs, GNN, RNNs or transformers. We find the two main reasons for modular architectures as being better performance on average and an easier to understand system that can be probed and tuned at component level. Another prominent trend is the extraction of object-centric information from inputs to enable object-level reasoning. This practice is frequently motivated by the desire for improved generalization and enhanced performance. However, while we explicitly resonate with the concept of object-level reasoning in general, we argue that

most of the contemporary physical reasoning benchmarks are still simple compared to the real world. This might make particularly the extraction of object information unrealistically easy and raises the question of how important the encoder component has to be in real world scenarios and how accurately this is represented by current benchmarks.

Creating benchmarks that do not offer shortcuts for machine learning approaches can pose a challenge. To address this, we propose a clear and formal description of the physical phenomena that should be tested. Additionally, we provide a comprehensive and distinct breakdown of all the components comprising a physical reasoning benchmark in Section 1. This approach aims to facilitate both the detection of possible shortcuts and a better understanding of the benchmark's underlying structure.

Regardless of whether language processing or interaction is involved, an orthogonal classification can be made based on the visual complexity of the benchmarks. Based on available solutions for the presented benchmarks, we conjecture that higher visual fidelity tends to correlate with higher benchmark difficulty if all other variables are kept fixed. However, depending on the task even a simplistic 2D task environment can already bring state-of-the-art agents to their limits. We argue that a generalist agent should solve both visually simple as well as complex tasks and that visually challenging benchmarks are necessary to achieve this goal. However, we perceive the visual complexity to be more or less orthogonal to the physical reasoning difficulty. Thus, one way to obtain a capable generalist agent could be a greater focus on curriculum learning w.r.t. not only the task difficulty but also the visuals. While we did not find any benchmark that provides this feature, we believe that smoothly ramping up input along with task complexity would be helpful in the creation of a generalist physical reasoning agent.

Drawing from our comprehensive benchmark comparison, we offer a taxonomy of physical reasoning benchmarks by grouping existing works and explaining the descriptive properties of each group. In addition, we aim to shed light on existing gaps in the current benchmark landscape. One such gap is the often-overlooked concept of relational physical variables, which is only explicitly implemented in the ComPhy and Physion++ benchmarks within our collection. While it is important to prioritize mastery of fundamental concepts, we believe that the advanced notion of relational physical variables deserves greater attention and should be incorporated into state-of-the-art physical reasoning benchmarks.

As the field continues to evolve, we anticipate the inclusion of new candidates to expand this collection of benchmarks. By regularly incorporating emerging benchmarks, we can ensure that our evaluation framework remains up-to-date and comprehensive.

The majority of datasets we present in this context are generated through simulation. While these datasets may offer a certain level of detail in the generated samples, they still fall short in terms of the complexity and noise found in real-world data. Notably, several benchmark papers have indicated or demonstrated that state-of-the-art models often struggle when confronted with real-world data. Related to this idea is the concept of domain randomization. In cases where an accurate mapping of the real world is challenging, domain randomization involves creating multiple instances of simulated domains with the hope that their combined characteristics encapsulate those of the real world. A well-known example of domain randomization is the OpenAI Rubik's Cube paper (Akkaya et al., 2019), which demonstrates how this technique can be applied effectively.

It is essential to acknowledge that there still is a substantial amount of unexplored terrain. In light of this, it is imperative for future benchmark authors to clearly define the physical concepts encompassed within their benchmarks, identify the types of physical variables that are relevant, and address the possibility of interactions with the environment. Future benchmarks can contribute to the continued advancement of our understanding of physical reasoning and pave the way for more comprehensive assessments of cognitive abilities across AI agents.

### Acknowledgments

This research was supported by the research training group "Dataninja" (Trustworthy AI for Seamless Problem Solving: Next Generation Intelligence Joins Robust Data Analysis) funded by the German federal state of North Rhine-Westphalia.

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
