# OpenReview forum: "Benchmarks for Physical Reasoning AI"
_TMLR — Accepted by TMLR_

### Review · Reviewer_bVmf · 2023-09-10

**Summary Of Contributions:**

This paper provides a survey of benchmarks for physical reasoning in AI. Sixteen benchmarks are discussed and taxonomized. The authors argue that an ensemble of benchmarks is most useful for measuring progress toward a generalist physical reasoning agent.

**Audience:**

Yes

**Claims And Evidence:**

Yes

**Requested Changes:**

I am not sure that it will be possible to address the concerns listed above within the scope of this submission. The high-level changes I think necessary include:
1. Establishing a stronger taxonomy and re-organizing the paper accordingly.
2. Providing a deeper analysis of the individual benchmarks.
3. Discussing the state-of-the-art for the benchmarks and physical reasoning more broadly.

### Minor
* I think the description of the solution for Figure 5 is wrong (“Solution: A circle is blocked vs. can be lifted”). According to the original work, it should be “Objects form a tower vs. an arch”.
* Make sure to check whether any of the figures copied from other works need explicit permission from the authors or journal.
* I found it odd to have a “Other Benchmarks” subsection that mentions two other benchmarks, when the main section includes so many already. Why not just add these two to the main?
* I don’t think robotic benchmarks need to be included in this paper, but I found this justification dubious: “However, we refrain from a deeper discussion of physical understanding for robots in this survey as well, since in robotics such understanding is frequently substituted by a suitable combination of embodiment and control algorithms to ensure that the robot’s movements remain compatible with the situations for which the robot is designed.”

**Strengths And Weaknesses:**

### Strengths
* The paper is thorough in the collection of benchmarks that it considers. I do not know of missing benchmarks that should obviously be included.
* The paper provides enough description of each benchmark that it is possible to understand the essential characteristics without referring directly to the benchmarks.
* Survey papers can be extremely useful -- our field needs more of them.
* I found Table 2, 3, and 4 to be quite useful, especially the input/output descriptions and the links to the benchmark repositories.

### Weaknesses
* Overall, my impression is that the survey is broad, but shallow. I think it would be more valuable if it were deeper. The bulk of the paper is a description of 16 benchmarks. Each description is relatively brief and does not offer much beyond what could be quickly understood from looking at the original work and glancing at its citations.
* I was also disappointed with the discussions of prior attempts to solve the benchmarks. I do not have a good impression of what the “state of the art” is for the benchmarks, nor a sense of which benchmarks are more or less difficult. Each benchmark does have a brief discussion of papers that have used the benchmark, but the discussion is too brief.
* I think it would also help to engage with the benchmarks a bit more critically. What are their strengths and weaknesses as benchmarks? Which of the many options are most worth considering in future research?
* Beyond the descriptions, some of the taxonomies proposed to organize the work are helpful, but they are not as crisp and organized as they could be.
   * Different aspects of physical reasoning are mentioned in paragraph 4 of the introduction, but these are not revisited.
   * I am not convinced that Global, External, Temporal, and Internal belong on the same spectrum, or that they can be “arranged from most pervasive to least conspicuous”. For example, gravity seems to be both “global” and “temporal”. Also, some object interactions (like a spring, or like sliding with friction) depend on both “external” and “internal” properties of the objects. I also don’t think that “external” and “internal” are the best words here. For “object variables visible from single frames”, I would suggest something like “immediate”; for “object variables exposed only through object interaction”; I would suggest something like “relational”. Finally, it’s worth noting that something like “position”, which is listed as an external property, is implicitly defined with respect to a world frame or some other object.
   * I didn’t glean much from Table 1 because it combines too many different axes of comparison. Some of these axes are later separated, but it’s hard to track the organization.
   * I like the idea to propose “clusters” of the benchmarks in Section 3, but I think the current attempt is not successful. First, “cluster” suggests that the benchmarks are going to be partitioned into groups, but this doesn’t seem to happen. It’s actually unclear what benchmarks belong to what clusters, and it certainly seems like some benchmarks are mentioned in multiple cluster discussions. The four clusters themselves also do not seem mutually exclusive, intuitively.
* I think survey papers are extremely valuable, but for a survey of benchmarks, I am not sure that a static paper is the best option. A “living document” (e.g., a GitHub page that accepts community contributions) could be far more valuable, since the landscape of benchmarks is still rapidly evolving. With that said, if the paper had a stronger analysis of the current state of the field, then it would be valuable to publish as a paper.

---

> ### Author Response · Authors · 2023-11-08
> **Response**
>
> >(1) Overall, my impression is that the survey is broad, but shallow. I think it would be more valuable if it were deeper. The bulk of the paper is a description of 16 benchmarks. Each description is relatively brief and does not offer much beyond what could be quickly understood from looking at the original work and glancing at its citations. (2) I was also disappointed with the discussions of prior attempts to solve the benchmarks. I do not have a good impression of what the “state of the art” is for the benchmarks, nor a sense of which benchmarks are more or less difficult. Each benchmark does have a brief discussion of papers that have used the benchmark, but the discussion is too brief. (3) I think it would also help to engage with the benchmarks a bit more critically. What are their strengths and weaknesses as benchmarks? Which of the many options are most worth considering in future research?
>
> We thank you for these suggestions, we've deepened our survey based on them. We added a new section containing information about contemporary benchmark solutions (Section 4) and list the state of the art approaches for each benchmark in its section. Furthermore, we added details about each benchmarks' strengths, notable contributions and discussed differences between the benchmarks where applicable.
>
> >(4) Beyond the descriptions, some of the taxonomies proposed to organize the work are helpful, but they are not as crisp and organized as they could be.
>
> We reorganised what are now Tables 1-3. We have also worked on improving our taxonomies, please see the responses below for details.
>
> >(4.1) Different aspects of physical reasoning are mentioned in paragraph 4 of the introduction, but these are not revisited.
>
> We adapted this part of the introduction and removed the aspects you refer to.
>
> >(4.2) I am not convinced that Global, External, Temporal, and Internal belong on the same spectrum, or that they can be “arranged from most pervasive to least conspicuous”. For example, gravity seems to be both “global” and “temporal”. Also, some object interactions (like a spring, or like sliding with friction) depend on both “external” and “internal” properties of the objects. I also don’t think that “external” and “internal” are the best words here. For “object variables visible from single frames”, I would suggest something like “immediate”; for “object variables exposed only through object interaction”; I would suggest something like “relational”. Finally, it’s worth noting that something like “position”, which is listed as an external property, is implicitly defined with respect to a world frame or some other object.
>
> Thank you for raising this point, we realise this paragraph was not very organised and we have adapted it. We have changed internal/external to immediate/relational since these words encapsulate the meaning of the variables. To address your points: a) We have made the reason for organising object variables on the same spectrum, and with increasing difficulty, clearer in the text. We agree that global variables cannot be put on the same spectrum for comparison. We have now put these as a separate category into the text.
>
> Regarding your point of physical variables not being clearly separated: We would like to clarify that this paragraph is about physical variables and not physical effects. Effects such as object interaction through a spring or sliding with friction are described by formulas, which in turn contain various physical variables of different types. We differentiate between object variables that differ from object to object, and environment variables that affect all objects. Gravity, for instance, is an environment rather than object variables (at least within our considered benchmarks). To uncover it, we agree that a temporal observation is necessary, but it is not a temporal object variable. We understand that this might have been misleading in the text and hope it becomes clearer by our separation of object variable types and global environment variables as a whole different category.
>
> >(4.3) I didn’t glean much from Table 1 because it combines too many different axes of comparison. Some of these axes are later separated, but it’s hard to track the organization.
>
> We restructured the Tables. We hope it is now easier to track the organization.

---

> > ### Author Response · Authors · 2023-11-08
> > **Response (continuation)**
> >
> > >(4.4) I like the idea to propose “clusters” of the benchmarks in Section 3, but I think the current attempt is not successful. First, “cluster” suggests that the benchmarks are going to be partitioned into groups, but this doesn’t seem to happen. It’s actually unclear what benchmarks belong to what clusters, and it certainly seems like some benchmarks are mentioned in multiple cluster discussions. The four clusters themselves also do not seem mutually exclusive, intuitively.
> >
> > Thank you for your comments.  We understand that the term "clusters" may be misleading since it typically implies mutual exclusivity. Therefore, we have revised our text to refer to these entities as "groups of benchmarks" instead. We believe this terminology choice avoids any implication of mutual exclusivity. In our updated text (Section 3.0), we clarify that the rationale behind forming these groups is rooted in the essential physical reasoning capabilities we consider crucial for a hypothetical generalist agent. Given that most benchmarks primarily focus on one of these specific abilities, we find it beneficial to put them into their respective groups for clarity and organization.
> >
> > >(5) I think survey papers are extremely valuable, but for a survey of benchmarks, I am not sure that a static paper is the best option. A “living document” (e.g., a GitHub page that accepts community contributions) could be far more valuable, since the landscape of benchmarks is still rapidly evolving. With that said, if the paper had a stronger analysis of the current state of the field, then it would be valuable to publish as a paper.
> >
> > We will link this survey to an "awesome"-like github repository with a list of Benchmarks for Physical Reasoning AI and make it open to pull requests, as well as actively maintain it up to date. Furthermore, we added a discussion of contemporary physical reasoning approaches to the paper (Section 4).
> >
> > >Requested Changes: I am not sure that it will be possible to address the concerns listed above within the scope of this submission. The high-level changes I think necessary include: Establishing a stronger taxonomy and re-organizing the paper accordingly. Providing a deeper analysis of the individual benchmarks. Discussing the state-of-the-art for the benchmarks and physical reasoning more broadly.
> >
> > We significantly updated the paper to address these concerns. We reorganised Tables 1-3 with improved taxonomy. Also, regarding the taxonomy see responses to 4, 4.1, 4.2, 4.3, 4.4 above. We added for each benchmark an overview of the state of the art solutions and a summary of the most important contributions of the benchmark (see paragraphs called "Solutions and related work" and "Contributions" in the updated manuscript). Additionally, we added Section 4 which discusses contemporary physical reasoning approaches more broadly.
> >
> >
> > >I think the description of the solution for Figure 5 is wrong (“Solution: A circle is blocked vs. can be lifted”). According to the original work, it should be “Objects form a tower vs. an arch”.
> >
> > Done.
> >
> > >Make sure to check whether any of the figures copied from other works need explicit permission from the authors or journal.
> >
> > Done.
> >
> > >I found it odd to have a “Other Benchmarks” subsection that mentions two other benchmarks, when the main section includes so many already. Why not just add these two to the main?
> >
> > We chose to create an "Other Benchmarks" subsection because the benchmarks included here are valuable for assessing the physical reasoning ability of agents under certain conditions but are testing too many confounding variables to be considered pure physical reasoning benchmarks. For the sake of completeness as a survey paper, we felt it was important to make readers aware of their existence while still emphasizing their limitations. To address this confusion, we have added an explanation to the start of the "Other Benchmarks" section.
> >
> > >I don’t think robotic benchmarks need to be included in this paper, but I found this justification dubious: “However, we refrain from a deeper discussion of physical understanding for robots in this survey as well, since in robotics such understanding is frequently substituted by a suitable combination of embodiment and control algorithms to ensure that the robot’s movements remain compatible with the situations for which the robot is designed.”
> >
> > We decided to exclude most robotics benchmarks from this survey, as robotics benchmarks often place more of a focus on competencies such as control, perception, navigation, manipulation, and localization, with physical reasoning simply being a potentially useful property. We have adapted the text in the introduction to make this clearer and hope this addresses your point.

---

> > > ### Comment · Reviewer_bVmf · 2023-11-09
> > > **Thanks for the revisions**
> > >
> > > Thank you for the very thorough revisions and replies to my comments. The paper looks significantly improved. Please follow through on the "awesome"-like github repository, I think that would be a valuable resource for the community.

---

### Review · Reviewer_deX2 · 2023-09-29

**Summary Of Contributions:**

This submission provides a survey of 16 different datasets and benchmarks for physical reasoning with AI.   Each of the 16 benchmarks is described, detailing the properties it has, the tasks in includes, and the approaches that have been used previously in the literature.  Gathering all of these benchmarks into a single place, along with the categorical descriptions of their properties, constitutes the main contribution of this submission.

A second contribution is the distillation from the benchmarks of 4 main abilities that the authors see as necessary to have a fully capable general physical reasoning AI.  The authors then cluster the 16 benchmarks based on which capabilities they include in their task.  This contributes a meaningful path forward to gradually increase the generalization ability of these AIs.  Researchers can test approaches across multiple benchmarks within a cluster.  As success is more broadly achieved within clusters, all 16 benchmarks can be approached with the same techniques, and success across the full set of benchmarks would be convincing that a general AI for physical reasoning tasks had been achieved.

Overall, this submission could be extremely helpful and impactful at guiding the direction of work of researchers in this area, as well as ensuring that evaluation of approaches can start to include multiple benchmarks.

**Audience:**

Yes

**Broader Impact Concerns:**

I have no broader impact concerns.

**Claims And Evidence:**

Yes

**Requested Changes:**

There are no changes that would be critical to securing my recommendation for acceptance.

Changes that I feel would simply strengthen the work:
- Page 1, last paragraph: "One part of presented . . . " should perhaps be "One part of the presented ..."
- Page 2, "taxonomy classification" sounds redundant to my ear.  I think that either of the words would convey the meaning by itself.
- Page 7, "The reward is the 1" should be "The reward is 1"
- Page 7, "...learns a state representative..." should this be "...learns a state representation..."?
- Page 7, "somewhat equivalent" might read better as simply "similar"
- Page 8, Figure 5 (and where it is referred to in the text).  The caption and text refer to a circle which is blocked or can be lifted.  However, I don't see any circle in the image.
- Page 13, second paragraph from the bottom: The sentences starting "The model is trained with videos . . ."  are not very clear.  I would encourage you to rework them.  The last sentence of the paragraph is long and hard to parse.
- Page 14, "is a worthwhile pursuing strategy" should be "is a worthwhile strategy to pursue"
- Page 14, First sentence of section 2.10.  The first citation (Girdhar and Ramanan, 2019) is awkward and should be modified, perhaps putting it is parentheses.
- Page 15, final paragraph.  There should be an "and" before the Sing et all (2022a) citation
- Page 16, There is a tiny paragraph that struck me as out of place, length-wise.  I might suggest joining it with either the preceding or succeeding paragraph.
- Page 21.  Directly before Section 2.15 there is an overleaf link that seems like it should not be there.
- Page 24, second to last line: "Agents and especially future ..." could be "Agents, especially future ..."
- Page 26, third paragraph from the top.  The sentence starting "The remaining two benchmarks involving classification . . ." doesn't make sense.  Specifically the word "however" seems out of place in the middle of the sentence.  I suggest reworking this.
- Page 26, second paragraph from bottom: "are usually no good..." could be "are usually not good..."
- Page 26, last line "searach" should be "search"

**Strengths And Weaknesses:**

Strengths:
- Overall the writing was quite clear and easy to follow.  The few exceptions will be noted later.
- The organization of the submission was clear
- The collection of benchmarks and related work is quite extensive and thorough.
- The characteristics collected and reported about each benchmark are very helpful.  I especially think that Tables 1-5  summarize these characteristics and attributes in a clear and useful way.
-Sections 3 and 4 do a good job of going beyond a simple survey of the benchmarks to also providing a roadmap for future research in this area.  I can envision this being very impactful to working on these problems.

Weaknesses:
- The overview of the different benchmarks at times started to feel repetitive.  But perhaps that is more due to the nature of the survey, rather than anything that the authors did wrong.

---

> ### Author Response · Authors · 2023-11-07
> **Response**
>
> Thank you for your feedback and valuable suggestions. We revised all points you raised as suggested.

---

### Review · Reviewer_ajqC · 2023-11-02

**Summary Of Contributions:**

The paper surveys a large number of existing physical reasoning benchmarks, describing them in detail and clustering along different axes.

**Audience:**

Yes

**Claims And Evidence:**

Yes

**Requested Changes:**

* The list of benchmarks at the introduction is currently a bit jarring. I think this foreshadows the clusters identified in section 3 but maybe this could be made more explicit. I'm not sure all the benchmarks need to be listed here. You could either name the clusters here and refer to section 3 or maybe instead add a table summarizing your clusters.
* Not necessarily a required change but I'm not sure the tags add a lot and make table 1 less self-contained.
* section 2.10 Girdhar & Ramanan -> (Girdhar & Ramanan)
* end of section 2.14: remove errant overleaf link?

**Strengths And Weaknesses:**

**Strengths:**
The paper surveys a large number of existing benchmarks, describing them in detail. The result is a very comprehensive overview of the state of the field which will be a great resource.

**Weaknesses:**
The paper is quite long in its current form and where possible could be edited down a bit. It has some minor formatting issues that should be addressed.

---

> ### Author Response · Authors · 2023-11-07
> **Response**
>
> >The paper is quite long in its current form and where possible could be edited down a bit. It has some minor formatting issues that should be addressed.
>
> We agree that the paper is long, but we suppose that the paper’s length is justified by its content. We designed every section of the paper as self-contained as possible so readers can easily jump between them and selectively read those they are interested in.
>
> >The list of benchmarks at the introduction is currently a bit jarring. I think this foreshadows the clusters identified in section 3 but maybe this could be made more explicit. I'm not sure all the benchmarks need to be listed here. You could either name the clusters here and refer to section 3 or maybe instead add a table summarizing your clusters.
>
> We removed the list and concentrated all information about benchmark groups in Section 3. Among other changes, we added Table 4 showing the group membership of the benchmarks there as well.
>
> >Not necessarily a required change but I'm not sure the tags add a lot and make table 1 less self-contained.
>
> We removed the old Table 1 and the tags altogether to replace them with two new informative and concise tables, Table 1 and Table 2. We moved discussions about benchmark clusters or groups to Section 3.

---

### Decision · Action_Editor_mF31 · 2023-11-23

**Recommendation:** Accept as is

**Comment:**

Good candidate for a survey certification, as it would be a good way for interested researchers to learn more about this particular sub-area.

**Audience:**

Yes, this article reviews a relatively popular sub-area and is self-contained enough to be of interest to folks well outside that area as well.

**Claims And Evidence:**

As accurate and clearly written review article, with several improvements since the initial submission. These led to clear approval from all reviewers and I agree.